Calcification response of planktic foraminifera to environmental change in the
Western Mediterranean Sea during the industrial era
Thibauld M. Béjard[1*], Andrés S. Rigual-Hernández[1], José A. Flores[1], Javier P.
Tarruella[1], Xavier Durrieu de Madron[2], Isabel Cacho[3], Neghar Haghipour[4], Aidan
Hunter[5], Francisco J. Sierro[1]
1. Area de Paleontología, Departamento de Geología, Universidad de Salamanca,
37008 Salamanca, Spain
2. Université de Perpignan Via Domitia, CNRS, CEFREM, Perpignan, France
3. GRC Geociències Marines, Departament de Dinàmica de la Terra i de l'Oceà,
Facultat de Ciències de la Terra, Universitat de Barcelona, Barcelona, Spain
4. Earth Sciences Department, ETH Zurich, Zurich, 8092, Switzerland
5. British Antarctic Survey, Natural Environment Research Council, Cambridge,
United Kingdom
*Corresponding author:  Area de Paleontología, Departamento de Geología,
Universidad de Salamanca, 37008, Salamanca, Spain. E-mail address:
thibauld.bejard@usal.es.

## 20  Abstract

The Mediterranean Sea sustains a rich and fragile ecosystem currently threatened
by multiple anthropogenic impacts that include, among others, warming, pollution
and changes in seawater carbonate speciation associated to increasing uptake of
atmospheric $CO_2$. This environmental change represents a major risk for marine
calcifiers such as planktonic foraminifera, key components of pelagic Mediterranean
ecosystems and major exporters of calcium carbonate to the sea floor, thereby
playing a major role in the marine carbon cycle. In this study, we investigate the
response of planktic foraminifera calcification in the northwestern Mediterranean
Sea on different time scales across the industrial era. This study is based on data
from a 12-year-long sediment trap record retrieved in the in the Gulf of Lions and
seabed sediment samples from the Gulf of Lions and the promontory of Menorca.
Three different planktic foraminifera species were selected based on their different
ecology and abundance: *Globigerina bulloides*, *Neogloboquadrina incompta,* and
*Globorotalia truncatulinoides*. A total of 273 samples were weighted in both sediment
trap and seabed samples
The results of our study show substantial different seasonal calcification patterns
across species: *G. bulloides* showed a slight calcification increase during the high
productivity period, while both *N. incompta* and *G. truncatulinoides* display a higher
calcification during the low productivity period. The comparison of these patterns
with environmental parameters revealed that Optimum Growth Conditions
temperature and carbonate system parameters are the most likely to influence
seasonal calcification in the Gulf of Lions. Interannual analysis suggest that both *G.*
*bulloides* and *N. incompta* did not significantly reduce their calcification between
1994 and 2005, while *G. truncatulinoides* exhibited a constant and pronounced
increase in its calcification that translated in an increase of 20% of its shell weight.
The comparison of these patterns with environmental data revealed that Optimum
Growth Conditions, Sea Surface Temperatures and carbonate system parameters
are the most likely parameters to influence calcification in the Gulf of Lions.
Finally, comparison between sediment trap data and seabed sediments allowed us
to assess the changes of planktic foraminifera calcification during the late Holocene,
including the preindustrial era. Several lines of evidence strongly indicate that
selective dissolution did not bias the results in any of our data sets. Our results
showed a clear calcification reduction between pre-industrial and post-industrial
Holocene and recent data with *G. truncatulinoides* experiencing the largest
calcification decrease (32-40%) followed by *N. incompta* (20-27%) and *G. bulloides*
(18-24%). Overall, our results provide evidence of clear reduction in planktic
foraminifera calcification in the Mediterranean most likely associated with ongoing
ocean acidification and consistent with previous observations in other settings of the
world's oceans.
**Key words:** Planktic foraminifera, foraminifera calcification, biogeochemical cycles,
Ocean Acidification, Mediterranean Sea.
1. Introduction
Growing population and its linked human activity since the industrial period (defined
according to Sabine et al., (2004) from 1800 and therein) has caused an increase in
carbon dioxide, which ecological and economic consequences are considered a
major threat (Ipcc, 2022). Atmospheric $CO_2$ concentrations during the Pleistocene
and Holocene ranged from 200 to 280 parts per million (ppm) (Loulergue et al., 2007;
Lüthi et al., 2008; Parrenin et al., 2007), but these values have increased
dramatically since the onset of the industrial period, exceeding the threshold of 400
ppm in 2015 for the first time for at least the last 800.000 years (Lüthi et al., 2008).
This increase is significantly more important since the 1950s, when rapid
atmospheric changes due to human activity took place, a process referred as "Great
Acceleration" (Head et al., 2022a). Since then, between, 25 and 30% of
anthropogenic $CO_2$ has been absorbed by the world's ocean (Sabine et al., 2004).
The ocean uptake of atmospheric $CO_2$ causes a drop in both pH and carbonate ion
concentration (Barker et al., 2012), lowering seawater alkalinity; this process is
commonly known as Ocean Acidification (OA), and it is expected to affect all areas
of the ocean and to have a wide impact on marine life (Hemleben et al., 1989). One
of the main questions about recent environmental change is how different
ecosystems and regions in global ocean are going to react to the ongoing increase
of anthropogenic atmospheric carbon dioxide.
A large body of evidence indicates that ocean acidification has substantial and
diverse effects on the distribution and fitness of a wide range of marine organisms
(Kroeker et al., 2013; Meier et al., 2014; Moy et al., 2009). For example, some fleshy
algae and diatom species have been shown to increase their growth and
photosynthetic activity at enhanced $CO_2$ concentrations (Kroeker et al., 2013). In
turn, most calcifying organisms such as calcifying algae, corals, pteropods,
coccolithophores and foraminifera are negatively affected by this process often
showing a reduction in their abundance, calcification and growth rates (Kroeker et
al., 2013; Orr et al., 2005).
Planktic foraminifera are a group of marine single-celled protozoans that produce
calcareous shells. Their distribution across the water column is conditioned by
factors that include, but are not limited to, food availability, temperature, salinity and
sunlight (Schiebel and Hemleben, 2005). These organisms are considered to play a
key role in marine carbon cycle and carbonate production, accounting for between
32 and 80% of the deep ocean calcite fluxes (Schiebel, 2002). Depending on their
ecology and feeding strategies, these organisms can be algal (dinoflagellates)
symbiont bearing or not symbiont bearing and be spinose or non-spinose. Planktic
foraminifera represent a useful tool for palaeoecological and palaeoceanographic
studies, as the abundances of different species and their geochemical signature
allow reconstructing sea surface temperatures and water column physical and
chemical properties (Lirer et al., 2014; Margaritelli, 2020; Schiebel and Hemleben,
106 2017).
Previous studies suggest that planktic foraminifera are sensitive to ocean
acidification (OA). Laboratory experiments indicate that when carbonate ion
concentration decreases, shell weight and calcification decrease too in a variety of
species (Bijma et al., 2002; Lombard et al., 2011). Species that host symbionts have
been described showing a higher tolerance to dissolution due to the capacity of algal
symbionts to alter immediate seawater chemistry (Lombard et al., 2009). Moy et al.
(2009) documented a decrease of 30-35% shell weight in the planktic foraminifera
*G. bulloides* during the industrial era in the subantarctic Southern Ocean, most likely
induced by anthropogenic-driven ocean acidification. A recent study by Fox et al.
(2020) showed that non-spinose (*N. dutertrei*) foraminifera species exhibit a more
pronounced calcification reduction than the spinose (*G. ruber*) species in response
to increasing $CO_2$. The main difficulty for studying the impact of OA on foraminifera
(and any calcifying organisms) resides in finding long-term continuous records in
order to be able to evaluate possible changes in shell calcification (Fox et al., 2020).
The Mediterranean Sea is a semi-enclosed sea with a high saturation state for calcite
(Álvarez et al., 2014). It is often considered as a "miniature ocean" and a "laboratory
basin" (Bergamasco and Malanotte-Rizzoli, 2010) which makes it a valuable zone to
study the response of marine calcifying organisms to environmental change.
In order to assess the impact of recent environmental change on planktic
foraminifera, in this work we present data from Planier sediment trap (data from 1993
to 2006) (Rigual-Hernández et al., 2012) and from seabed sediments from three
different sites located in both the Gulf of Lions and the promontory of Menorca. The
advantage of sediment traps is that they can provide data coming from annual fluxes,
avoiding the effects of seasonal abundance and ontogeny and making interannual
comparisons more reliable (Jonkers et al., 2019). Three different planktic
foraminifera species, each of which characterized by contrastingly different depth
habitats and ecologies, were selected for our analysis: *G. bulloides,* a spinose
opportunist surface dweller that lies above the thermocline (Schiebel and Hemleben,
2014); *N. incompta,* a non-spinose temperate surface dweller; and *G.
truncatulinoides,* a non-spinose deep dwelling species which migrates through the
water column with a complex life cycle. Our aims for this study are: (i) to compare
two widely used foraminifera weighing and size-normalization techniques and
provide a baseline of modern foraminifera weight data and calcification in the
Western Mediterranean against which future changes in foraminifera calcification
can be assessed (ii) document seasonal and interannual trends in the planktic
foraminifera calcification of the three planktic foraminifera species, and (iii) evaluate
possible changes in shell calcification through the Holocene to the present day by
comparing shell weights of the foraminifera collected by the traps with those of the
seafloor sediments.

## 2. Study area

The Mediterranean is a semi-enclosed sea and it is considered a concentration basin
(Bethoux et al., 1999) with a negative hydrological budget: fresh water inputs do not
compensate the overall basin evaporation. The surface oceanic waters that enter
the Mediterranean through the Strait of Gibraltar and spread towards the eastern
basin compensate this negative balance. The Corsica Channel represents the choke
point for water circulation in the Northwestern Mediterranean (Millot, 1999) through

which waters of the Tyrrhenian flow into the Ligurian Sea, where the Northern Current (NC) is formed. The NC largely controls the circulation all over the western and northwestern part of the Mediterranean Sea, including the Gulf of Lions (Millot, 1991) and the Balearic Sea (Fig. 1a).

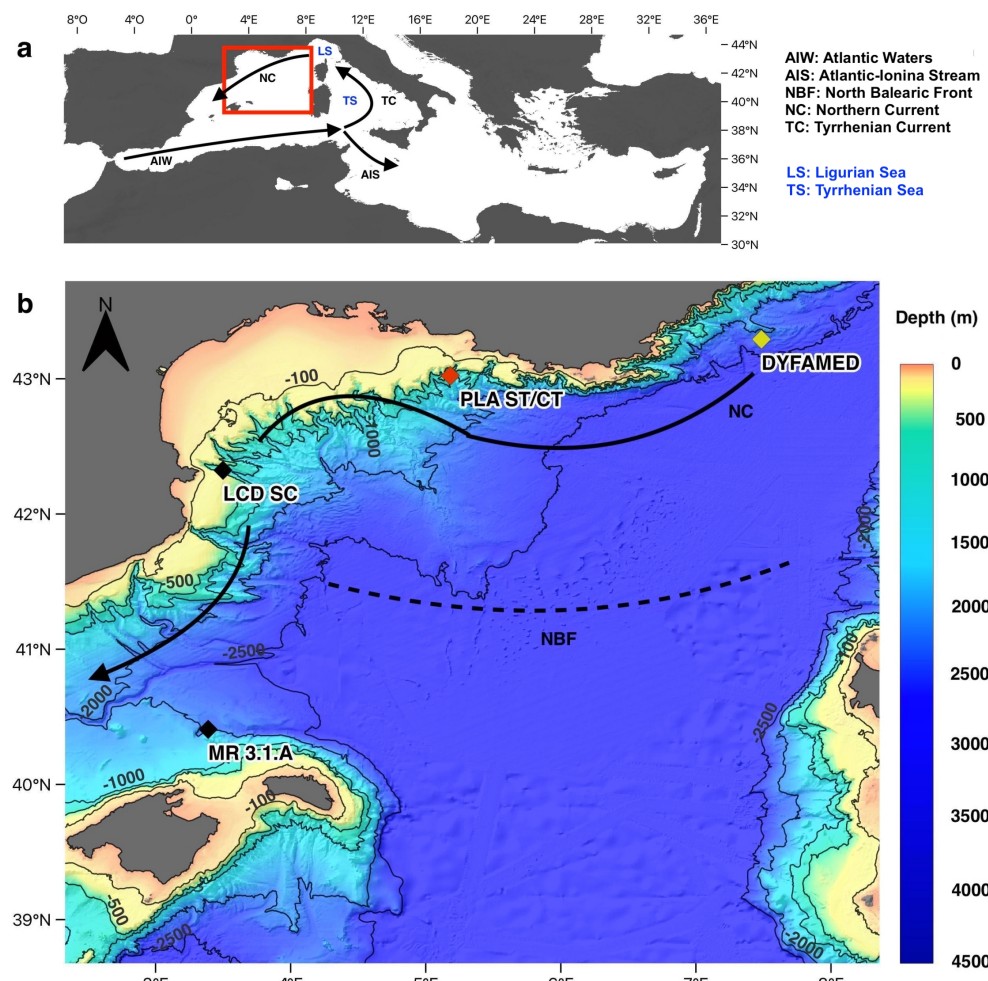

**Figure 1. a.** Study area location in the Mediterranean Sea and general surface circulation **b.** geographic setting of the Gulf of Lions and location of studied sites. Red diamond shows the position of the Planier site sediment trap and core-top (PLA ST/CT). Black diamonds represent the location of the seabed sediments samples analyzed from Lacaze Duthiers canyon (LCD SC) and Menorca promontory (MR 3.1A). Yellow diamond represents the location of the Dynamics of Atmospheric Fluxes in the MEDiterranean Sea (DYFAMED) site, located 200 km upstream Planier station position. Arrows represent the surface circulation (Millot, 1999). The topographic model was downloaded from the GEBCO database.

Moreover, the Mediterranean is recognized as a sensitive region to increasing atmospheric $CO_2$ (Ziveri, 2012) due to the fast turnover time of its waters (Béthoux

et al., 2005) and the fast penetration of anthropogenic $CO_2$ (Schneider et al., 2007).
Sea surface temperatures are predicted to increase by 1.5-2°C by the end of the
century, a faster rate than the global average (Lazzari et al., 2014). pH is expected
to decrease according to the global average (0.3-0.4 units by 2100) or even exceed
the global trend (Hassoun et al., 2015). The Mediterranean Sea is also affected by
other stressors, which impact marine organisms in many ways (Lejeusne et al.,
2009). Finally, it is also a region shaped by human development and its associated
activities interact with environmental changes (Mediterranean Experts on Climate
Change, MedECC, 2019).
The Gulf of Lions is located in the northwestern part of the Mediterranean Sea, and
its morphology presents a continental slope with an array of complex submarine
canyons (Rigual-Hernández et al., 2012) (Fig. 1b).
Vertical mixing, generated by intense surface cooling and evaporation, occurs in
winter in the Gulf of Lions driven by cold, dry northern winds, resulting in dense water
on the shelf and offshore (Durrieu de Madron et al., 2005; Houpert et al., 2016; Millot,
1991). This winter mixing recharges surface waters with nutrients. This enrichment
with increased solar radiation stimulates primary production in spring. Increasing
heat fluxes during spring and summer cause water mass stratification and nutrient
depletion, which lasts until late summer, until fall cooling breaks the stratification of
the water column and causes a fall bloom (Heussner et al., 2006; Monaco et al.,
1999; Rigual-Hernández et al., 2012). River inputs are the main source of suspended
particles in the Gulf of Lions, and the Rhone river represents the most important river
in the northwestern Mediterranean; however, other sources include Saharan dust
deposits and biological production (Heussner et al., 2006; Monaco et al., 1999).
Overall, the oceanographic setting of the Gulf of Lions is an exception to the general
oligotrophy of the Mediterranean Sea.
## 3. Material and methods
### 3.1. Sediment traps, core-tops and sediment cores.
A series of deployments of sediment traps mooring lines in the Gulf of Lions
continental margin was initiated in 1993 within the framework of several French and
European projects (PNEC, Euromarge-NB, MTP II-MATER, EUROSTRATAFORM)
and the monitoring of two sites, Planier and Lacaze-Duthiers stations (Fig. 1),
continues as a component of the MOOSE program (Mediterranean Ocean
Observing System for the Environment) (Coppola et al., 2019). Planier station
(43°02'N, 5°18'E) is located at the northeastern end of the Gulf of Lions, in the axis
of the Planier Canyon. The sediment trap used for this work was located at around
530 m water depth in a water column of ~1000 m. Further details of the mooring
design can be found in Heussner et al., (2006). Planktic foraminifera fluxes for the
1993 to 2006 period were documented by Rigual-Hernández et al., (2012). Here, we
used the samples from the latter study for our weight and calcification analysis. This
sediment trap is used here as a baseline of the planktic foraminifera dwelling in the
modern Mediterranean Sea. Moreover, we analyzed a set of core top and sediment
cores collected from several locations of the Northwestern Mediterranean that are
considered to represent foraminifera assemblages sedimented during the Holocene
era (Table 1).
**Table 1**. Description of the core tops used in this study. Data for Planier core-
top (PLA CT) and Lacaze-Duthiers sediment core (LCD SC) are available in
Heussner et al., (2006), and data concerning Menorca sediment core (MR
3.1.A) can be found in Cisneros et al., (2016). Conventional [14]C ages, 1-sigma
uncertainties, local reservoir and the calibrated age have been rounded
according to convention.

| Site | Location | Water depth (m) | Sediment Samples | Samples Depth (cm) | Sample dated | Species dated | Radiocarbon age ([14]C years BP) | 1-sigma error ([14]C years ) | Local reservoir ([14]C years BP) | Calibrated age (cal. years BP) |
|---|---|---|---|---|---|---|---|---|---|---|
| **PLA CT** | 42.989° N 5.121° E | 1095 | 2 | 0-1 | 0.5-1 cm | *G. bulloides* | 490 | 60 | 165 ± 93 | Out of range |
| **LCD SC** | 42.265°N 3.54°E | 990 | 7 | 0-5 | 0.5-1 cm | *G. bulloides* | 460 | 60 | 165 ± 93 | Out of range |
| **MR 3.1.A** | 40.29°N 3.37° E | 2117 | 40 | 0-27 | 14-14.5 cm | *G. bulloides* | 1980 | 65 | 165 ± 93 | 1555 |


## 3.2. Sediment core samples processing

A total of 2 sediment samples from Planier core top, 7 from Lacaze-Duthiers
sediment core and 40 from Minorca sediment core were weighed (Table 1). Dry bulk
sediment samples from all sites was weighed using a Sartorius CP124S balance
(precision= 0.1mg).
The samples were then wet-sieved in order to separate the <63 $\mu$m fraction and dry
sieved to separate the bigger fractions (>150 $\mu$m and >300 $\mu$m). The sediment
washing was carried out with potassium phosphate-buffered solution (pH= 7.5) in
order to optimize foraminifera preservation. Each fraction was oven dried at a
constant temperature (50°C) and then weighed. The >150 $\mu$m fraction was used for
identification, counting and shell morphometric and weight analyses.

## 3.3. Ecology and life cycle of *Globigerina bulloides*, *Neogloboquadrina incompta* and *Globorotalia truncatulinoides*

*G. bulloides* is a spinose surface to sub-surface dweller (Schiebel and Hemleben,
2017), found in the upper 60 m of the water column. This species has affinity for
temperate to sub-polar waters and upwelling systems in lower to mid latitudes
(Azibeiro et al., 2023; Bé et al., 1977). In terms of its seasonal distribution, it has
been documented associated to  enhanced productivity periods in mid to high
latitudes (Chapman, 2010; Schiebel and Hemleben, 2005). No symbiont algae are
hosted by this species and, contrary to most spinose species, its diet is mainly algae
based (Schiebel et al., 2001). *G. bulloides* shows an opportunistic feeding and
strategy, leading to a high abundance in the foraminifera assemblages preserved in
the sedimentary record. This is despite its tests have been documented to be more
susceptible to dissolution than the average of the planktic foraminifera species
(Dittert et al., 1999).
*N. incompta* is a surface dweller abundant in subpolar to temperate water masses
across all the ocean basins (Kuroyanagi and Kawahata, 2004). This is a non-spinose
species and does not carry symbiont algae. In North-Atlantic waters, *N. incompta* is
a major component of foraminifera assemblages from late spring to late fall, and
generally, is the dominant foraminifera species during late summer when maximum
shoaling of mixed layer depths occur (Schiebel and Hemleben, 2000). It shows a
minor presence in low latitudes and during periods of enhanced nutrient supply, *N.*
*incompta* is outnumbered by other more opportunistic species, (Schiebel et al.,
2002).

*G. truncatulinoides* is considered the deepest dweller among the extant planktic
foraminifera, with living specimens documented below 2000 m (Schiebel and
Hemleben, 2005). Considered a widespread species, it can be found from subpolar
to subtropical water masses (Schiebel and Hemleben, 2017). It is a non-spinose
species, and it does not carry any symbiont algae (Margaritelli, 2020). An important
aspect to highlight about this species is its complex life cycle. reproduces once a
year in the upper water column during late winter, when mixing of the water column
allowed the migration of juveniles to the surface waters (Lohmann and Schweitzer,
1990; Schiebel et al., 2002). The former authors speculated that nutrient availability
and the avoidance strategies to predation could explain this its life cycle. Then, the
adult migrate downward the water column and spend the rest of their life cycle
developing an additional calcite layer in cooler waters below the thermocline
(Lohmann and Schweitzer, 1990; Wilke et al., 2009). Around 70% of *G.*
*truncatulinoides* calcification has been estimated to take place at around the
thermocline, while the remaining 30% take place in surface waters (LeGrande et al.,
2004).


**3.4.    Foraminifera picking and mass and size estimations**
Different sizes were selected depending on the maximum availability of each
species: 250-300, 200-250 and 400-500 *µ*m for *G. bulloides, N. incompta* and *G.*
*truncatulinoides*, respectively. For the latter species, both coiling morphotypes were
selected although the right coiling was substantially less abundant representing less
than 3% in our counts, a feature consistent with the literature that indicates a low
presence of right coiled specimens (Margaritelli et al., 2020; 2022).
A total of 273 foraminifera samples were picked for this study, 126 coming from the
sediment trap and 147 from the three sediment cores and core tops (Table 2).
However, these numbers represent the total of samples analyzed but unique
samples number is lower, as not all the sediment trap samples presented the three
species in high enough numbers to perform the picking. The species were analyzed
in size fractions in order to estimate the efficiency of sieve fractions and the impact
of size and morphometric parameters on the foraminifera weight and calcification,
The mean weight of each available sediment trap sample was obtained by weighting
between 15 to 45 specimens of *G. bulloides* (mean N= 27), 5 to 25 *N. incompta*
(mean N= 15) and 5 to 25 *G. truncatulinoides* (mean N= 13). Concerning the
analyses of the core top and sediment core samples, between 15 and 25 *G. bulloides*
and *N. incompta* (mean N= 20 for both) and between 9 and 25 *G. truncatulinoides*
(mean N= 18) were picked.
Each foraminifera sample was then cleaned by gentle ultrasonication (50 Hz) for 5
to 75 seconds (depending on the species and the degree of visual uncleanliness) in
methanol in order to clean the shells. The samples were then left to dry in a
temperature-controlled oven at 50°C. One out of three analyzed samples were
weighted before and after cleaning in order to assess potential shell mass losses
and shell preservation due to ultrasonication. Our results indicate that this method
has little impact on shell preservation with around 95% of the total foraminifera
conserved in good conditions. Weight loss between non-cleaned samples and
cleaned samples are a mean 0.5 to 3 $\mu$g depending on the species, mainly due to
the presence of clay and non-calcite material in the shells, which justifies this
cleaning process (see Supplementary fig. 6).
The weightings were carried in the micropaleontology laboratory of the Geology
Department at University of Salamanca using a Sartorius ME5 balance (precision=
0.001 mg). This method allowed us to obtain foraminifera Sieve Based Weight
(SBW) by dividing the average shell weight per sample (5-45 tests) by the total
number of foraminifera within each sample. The lowest number of individuals
selected per sample was five in order to maximize the number of samples available
for our study. According to Beer et al., (2010), the higher the number of individuals,
the more reliable SBWs are. Here we aim to compare SBW results with a measured
weight technique. Measured techniques are acknowledged to be reliable with a lower
number of individuals, therefore a minimum of five individuals were selected in order
to compare the two techniques.
However, it has been described traditionally used sieve fractions method is
considered unreliable because of the effect of morphometric parameters on the
foraminifera weight (Beer et al., 2010). In order to remove the size effect on the
weight, the mean SBW was normalized to the mean diameter and area of the
planktic foraminifera to obtain Measurement Based Weights (MBW). Morphometric
parameters were measured using a Nikon SMZ18 stereomicroscope equipped with
a Nikon DS-Fi3 camera and NISElements software. These measurements were
carried out on the same shells that were weighted. Foraminifera shells were
positioned in order to obtain the maximum area of each individual, in this case, the
umbilical side (aperture facing upwards) was measured for the three species.
$MBW_{area}$ and $MBW_{diameter}$ were calculated according to the following formula
(Aldridge et al., 2012; Beer et al., 2010), where "parameter" accounts for "area" or
"diameter":

$$MBW\,sample = \frac{mean\,SBW_{sample} \times mean\,parameter_{size\,fraction}}{mean\,parameter_{sample}}$$

"Size fraction" accounts for the mean of the parameter (area or diameter) measured
in all the sites studied, while "sample" accounts for the mean of the parameter in the
particular sample being measured. The advantage of these measurements is that
the resulting MBW is being given with a weight unity ($\mu$g), thereby allowing direct
comparison with other studies (Beer et al., 2010) and useful for estimating their
contributions to marine biogeochemical cycles.
Correlations between SBW and $MBW_{area}$ against area are displayed in Fig. 2. The
reason for this comparison is to show the relation between size and weight. In order
to avoid the impact of having the bigger specimens displaying the heaviest weight
and impacting the mean weight (therefore calcification indicator) of the sample.
Finally, in order to compare weights patterns from the sediment trap with weights
from core tops and sediment cores and overcome the seasonality effect, MBWs were
flux-weighted. Mean monthly MBWs values from each species were multiplied by
the corresponding mean monthly flux and then divided by the total annual flux of the
corresponding species. For these calculations, the flux data from each species
estimated for the >150 $\mu$m fraction from Rigual-Hernández et al., (2012) was
employed.

**3.5.   Environmental data**

Foraminifera fluxes and abundances together with chlorophyll-*a* were taken from Rigual-Hernández et al., (2012) for the entire time span of the analyzed samples. Both fluxes and abundance come from direct sediment observation from the Planier site, while chlorophyll-*a* data was obtained from SeaWiFS monthly measurements through NASA's Giovanni program on a 0.2 x 0.2° area around the mooring location. SeaWiFS measurements started in 1997 and were used due to the lack of direct chlorophyll measurements in our samples. Sea Surface Temperature (SST) was recovered from the NOAA database with the same gridding as the data from the NASA's Giovanni program.

Salinity, nutrient concentrations (nitrates and phosphates) and carbonate system parameters data were collected from the DYFAMED database (http://www.obs-vlfr.fr/dyfBase/index.php) (Coppola et al., 2008; 2021). DYFAMED site is located around 200-220 km (Fig. 1b) east of the sediment trap location (43°25′N, 7°52′E), in the Ligurian Sea. From an oceanographic view, its situation is upstream of the NC circulation and can be considered representative of seasonal and interannual variability of biological and water column properties of the open-ocean waters in the northwestern Mediterranean (Heussner et al., 2006; Meier et al., 2014). Alkalinity and total carbon measurements were available for years 1998 to 2000 and mid 2003 to 2005. Missing values comprised in these years were replaced with values obtained from linear regression of the measurements from above and below. The CO2SYS macro has been used to reconstruct the $[CO_2]$, $[CO_3^{2-}]$, $[HCO_3^-]$ and pH values from the measured total alkalinity and dissolved inorganic carbon. The constants used were the $CO_2$ dissociation constant by Mehrbach et al., (1973) refit by Dickson and Millero, (1987); the KHSO4 by Dickson, (1990); and the seawater scale for pH.

### 3.6. Statistical analysis

In order to have uninterrupted monthly environmental values from the DYFAMED site during available measurements, a resampling every 10 days has been carried out with the QAnalySeries program.

Independence and correlation between the area the different species SBWs and MBW$_{area}$ was tested using a Pearson linear correlation test with an R script (see Supplementary material).

Seasonal correlation analyses were carried out with the Statistica program. A $p<0.05$ was used in order to consider a correlation as significant. The number (N) of correlations depended on data availability and was 10 for *G. bulloides*, 9 for *N. incompta* and 12 for *G. truncatulinoides*.

It should be noted that the analysis of interannual trends was hindered by gaps in the sediment trap record and by the low number of specimens during some sampling

intervals. Therefore, interannual trends in planktonic foraminifera calcification should
be interpreted with caution.
The influence of a suite of environmental variables upon MBW$_{area}$ was assessed
using General Additive Models (GAM) (*mgcv* R-package). For each species, MBW
was modelled as a smooth function of each environmental variable (see
Supplementary figs. 3, 4 and 5). The limited number of data points restricted GAM
complexity to a single explanatory variable, so environmental interaction effects were
not assessed.
In order to investigate the difference between the MBW data from the sediment trap
and the core-top /sediment cores, a non-parametric two-way Mann-Whitney test has
been performed. The aim of this test is to analyze the difference between the median
of the different datasets. A p-value <0.05 has been used to consider the median of
two datasets different.
### 3.7.   Radiocarbon dating
Between 50-100 individuals of well-preserved *G. bulloides* shells (>150 $\mu$m) were
picked for radiocarbon analyses. Radiocarbon ($^{14}C/^{12}C$) was measured as $CO_2$ with
a gas ion source in a Mini Carbon Dating System (MICADAS) at the Laboratory of
Ion Beam Physics from ETH Zürich. The employed automated method consists of
initial leaching of the outer shell to remove surface material with 100 $\mu$l of ultrapure
HCl (0.02M) and the subsequent acid digestion of the remaining carbonates with
100 $\mu$l of ultrapure $H_3PO_4$ (85%) (Wacker et al., 2013). Therefore, no cleaning was
applied after the picking contrary to the samples used for mass and size
measurements. Marble (IAEA-C1) was used for blank correction and results were
corrected for isotopic fractionation via $^{13}C/^{12}C$ isotopic ratios.
Conventional radiocarbon age for sample 14-14.5cm from MR 3.1.A site was
calibrated with the on-line calibration program CALIB (Stuiver and Reimer, 1993)
using the Marine20 curve, which applied a marine reservoir correction of 550 $^{14}C$
years (Heaton et al., 2020) to the corresponding $^{14}C$ age and error. Additionally, a
local reservoir effect (Stuiver and Braziunas, 1993) of -165 ± 95 $^{14}C$ years was
considered. This local reservoir was calculated as the average of the 8 nearest
points to the sample location from the Marine Reservoir Correction database
(Reimer and Reimer, 2001), whose values have already been corrected for the
Marine20 curve. $^{14}C$ ages from samples 0.5-1cm from both PLA CT and LCD SC
lied out of the range for calendar calibration, implying these samples contain some
bomb $^{14}C$ and cannot be considered pre-industrial. In order to have an estimation of
the time span that could be covered by these dates, the same marine and local
reservoir corrections were applied to the most recent $^{14}C$ date that could be corrected
(i.e. 603 $^{14}C$ years BP). As the $F^{14}C$ for this sample was <1 (see Supplementary
table 1), this means that the $^{14}$C found in these samples is not dominated by the
bomb carbon. Here we propose a 110-50 cal. years BP age for these samples. Then,
these samples are considered post-industrial. The detailed results of the calibration
and the $^{14}$C dating can be found in the Supplementary figs. 1 and 2.
Finally, it is important to consider that these $^{14}$C ages represent mean average
values. Therefore, time integration within each sample and the effects of bioturbation
could cause a variation on the foraminifera real ages (Dolman et al., 2021).
Both the samples and dates obtained are detailed in Table 1. Planktic foraminifera
present in the dated samples that were not selected for radiocarbon dating were also
analyzed following the methodology described previously.

## 441   4. Results


### 443   4.1. Shell morphometric parameters and shell-weight normalization

Table 2 shows the results of the shell area, diameter and SBW, the total foraminifera
samples analyzed, the mean number of individuals per sample and the total
individuals measured at each of the studied sites.

**Table 2**. Minimum, mean, maximum and standard deviation values of shell
area, diameter and SBW for *G. bulloides, N. incompta* and *G. truncatulinoides*
at all studied sites. The last 3 columns show the number of samples, the mean
number (N) of individuals analyzed per sample and total number of individuals
measured for each site.

| PLA Sediment Trap | Area (μm²) | | | | Diameter (μm) | | | | Sieve Based Weight (SBW, μg) | | | | Total samples | N per sample | total N |
|---|---|---|---|---|---|---|---|---|---|---|---|---|---|---|---|
| | Min | Mean | Max | Std.Dev | Min | Mean | Max | Std.Dev | Min | Mean | Max | Std.Dev | | | |
| *G. bulloides* | 16978 | 57353 | 168492 | 17261 | 147.0 | 267.5 | 463.2 | 38.6 | 3.21 | 4.43 | 5.60 | 0.66 | 35 | 27.2 | 893 |
| *N. incompta* | 26234 | 42821 | 135422 | 8934 | 182.8 | 232.4 | 415.2 | 22.6 | 3.17 | 4.45 | 5.40 | 0.59 | 32 | 15.0 | 455 |
| *G. truncatulinoides* | 70712 | 178952 | 527622 | 63572 | 291.9 | 468.5 | 819.6 | 81.9 | 10.67 | 23.11 | 39.57 | 7.79 | 59 | 13.0 | 729 |
| PLA Core-Top | | | | | | | | | | | | | | | |
| *G. bulloides* | 37163 | 55395 | 87894 | 12302 | 217.5 | 264.0 | 334.5 | 28.8 | 5.00 | 5.22 | 5.43 | 0.30 | 2 | 17.3 | 39 |
| *N. incompta* | 27635 | 36927 | 49619 | 5447 | 187.6 | 216.3 | 251.4 | 15.9 | 4.46 | 4.46 | 4.46 | 0.00 | 2 | 19.7 | 41 |
| *G. truncatulinoides* | 89778 | 174748 | 233229 | 44313 | 338.1 | 467.7 | 544.9 | 61.9 | 34.80 | 35.40 | 35.90 | 0.70 | 2 | 14.7 | 34 |
| MIN Sediment core | | | | | | | | | | | | | | | |
| *G. bulloides* | 20895 | 52132 | 138424 | 8722 | 163.1 | 256.8 | 419.8 | 20.5 | 4.00 | 5.07 | 6.57 | 0.46 | 40 | 19.6 | 761 |
| *N. incompta* | 24003 | 35098 | 57264 | 4658 | 174.8 | 211.0 | 270.0 | 13.7 | 3.45 | 4.11 | 5.00 | 0.34 | 40 | 20.3 | 791 |
| *G. truncatulinoides* | 116686 | 166318 | 365851 | 23262 | 385.4 | 459.1 | 682.5 | 30.8 | 28.33 | 34.99 | 42.60 | 3.25 | 40 | 14.4 | 576 |
| LCD Sediment core | | | | | | | | | | | | | | | |
| *G. bulloides* | 27624 | 52472 | 116605 | 8793 | 187.5 | 257.7 | 385.3 | 20.4 | 4.35 | 4.73 | 5.19 | 0.31 | 7 | 20.1 | 136 |
| *N. incompta* | 28089 | 37789 | 51284 | 4972 | 189.1 | 218.9 | 255.5 | 14.4 | 3.68 | 4.12 | 4.50 | 0.26 | 7 | 19.8 | 134 |
| *G. truncatulinoides* | 82534 | 143138 | 393754 | 41620 | 324.2 | 423.3 | 708.1 | 55.9 | 25.27 | 26.68 | 30.66 | 1.94 | 7 | 15.3 | 105 |

Overall, the mean values for both diameter and area correspond to mean narrowed size fraction used during the picking, but morphometric parameters show some variability between the studied sites. Standard deviation of both area and diameter values for the three species are higher in the sediment trap record than in seafloor sediments, with mean values (of all three species) of 82% higher for area and 69% higher for diameter. SBW exhibits the same pattern as both area and diameter standard deviation is a mean 130% higher in the Planier sediment trap. Regarding the variability across the seafloor samples, Planier core-top exhibits a greater area and diameter values (about 40 to 50% increase for the three species) compared to those of the other two sediment cores, probably due to the fewer samples analyzed (Table 2).

The Planier sediment trap results (Table 2) show a higher standard deviation for both area and diameter for the three species, i.e. 76 % and 68% higher for *G. bulloides* compared to the data from core tops, 78% and 54% for *N. incompta* and 81% and 73% for *G. truncatulinoides*.

Because of the lack of precision of the initial individuals picking, carried out with a micrometer installed in the microscope, the selection is not totally accurate. Due to this issue, one third of the of the total measured foraminifera (i.e. 1645 of 4694) were out of the desired size fraction, of which 12% were bigger (580/4694) and 23% were smaller (1065/4694). Nonetheless, only 0.02% were more than 20% out of the selected size range (64/4694 more than 20% bigger and 29/4694 more than 20% smaller). Mean size difference for the foraminifera out of the size fraction is around 7%. Results vary according to the site and the species. 50% of the individuals from the Planier sediment trap (1046/2077) and 26% of the individuals coming from the

core tops (692/2617) were out of range. *G. bulloides* showed a 45.5% (53.2% in the
sediment trap and 39.3% in the core-tops samples) of individuals out of selected size
fraction, while this value is 21.5% (22.2% in sediment trap, 21.1% in sediment cores)
for *N. incompta* and 35% for *G. truncatulinoides* (53.4% in sediment trap, 16.7% in
sediment cores).

Even though a narrow size class was selected for each species (see section 3.4), a
clear influence of the area on the SBW was found in our data set (Fig. 2).

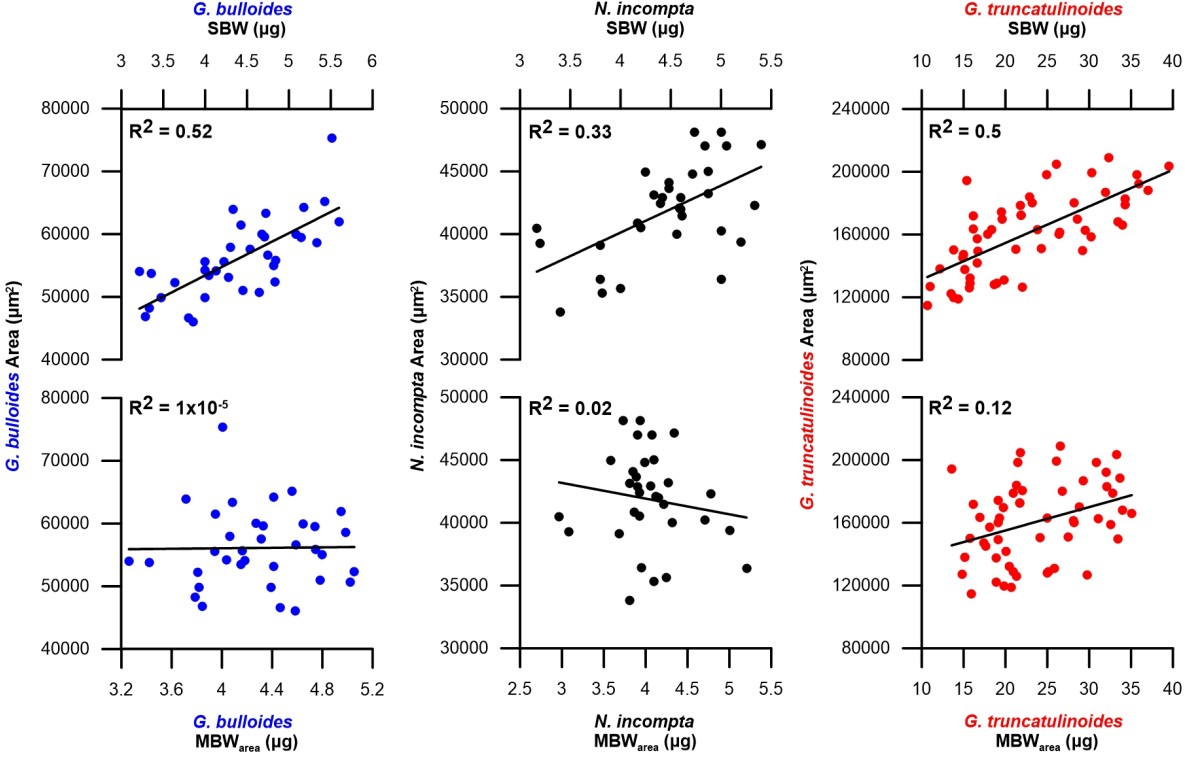


**Figure 2**. SBW in $\mu$g and MBW$_{area}$ in $\mu$g against the mean test area in $\mu$m$^2$ for
foraminifera samples in the Planier sediment trap. Dark blue dots correspond
to *G. bulloides*, black dots *to N. incompta* and red dots to *G. truncatulinoides*.
In Fig. 2, we show the values for the Planier sediment trap record for both SBW and
MBW$_{area}$ for all three species considered in this study, and their r$^2$ with the measured
area. In particular, SBW shows a positive correlation with area: $0.33 < r^2 < 0.53$. This
indicates that the SBW is dependent on the size of the specimens within the selected
size range. Thus, to isolate the component of variation in foraminifera shell thickness
that represents a change in calcification and does not occur as a direct result of
changes in shell size, normalization of the shell weight was performed following the
formula detailed in section 3.3. (Beer et al., 2010). After normalization MBW$_{area}$
shows no correlations with area: $1 \times 10^{-5} < r^2 < 0.12$ (Fig. 2). Note that the weight
variations in our dataset are quite considerable, especially for *G. truncatulinoides*,
probably due to the wider size fraction. Diameter does show correlation with SBW:
$0.33 < r^2 < 0.5$; and shows a non-negligible correlation with $MBW_{diam}$: $0.2 < r^2 < 033$. Our
data demonstrates that SBW correlates more strongly with $MBW_{diam}$ than with
$MBW_{area}$ for the 3 species: $0.9 > 0.48$ for *G. bulloides*, $0.89 > 0.52$ for *N. incompta* and
$0.97 > 0.81$ for *G. truncatulinoides*. These values are consistent with previous studies
(Beer et al., 2010).
**Table 3.** Pearson correlation test results for the three species correlation
between area ($\mu m^2$) and both SBW ($\mu g$) and $MBW_{area}$ ($\mu g$). Here c.i. stands
for "confidence interval". Significant r values (0<c.i.<1) are set in bold.

| | Area ($\mu m^2$) | | | | | |
|---|---|---|---|---|---|---|
| | *G. bulloides* | | *N. incompta* | | *G. truncatulinoides* | |
| | r | c.i. | r | c.i. | r | c.i. |
| **SBW ($\mu g$)** | **0.72** | 0.52, 0.85 | **0.57** | 0.28, 0.77 | **0.62** | 0.41, 0.76 |
| **MBW$_{area}$ ($\mu g$)** | 0.014 | -0.32, 0.35 | -0.15 | -0.47, 0.21 | 0.21 | -0.09, 0.44 |

Furthermore, a Pearson correlation test (see section 3.6) has been carried out in
order to assess the influence of area on SBW and $MBW_{area}$ (Table 3). Results
showed that the SBWs from the three species correlated positively and significantly
(0< c.i <1). with their corresponding areas (0.57< r <0.72). Concerning the MBWs,
no significant (0> c.i. >1) correlations with the area are observed (-0.15< r <0.2).
Therefore, these correlations further highlight the fact that SBW values are
significantly influenced by shell area, while $MBW_{area}$ values appeared to be
independent of the area.
Differences between SBW and both $MBW_{area}$ vary depending on the species: SBW
is slightly heavier for *G. bulloides*, heavier for *N. incompta* and lighter for *G.
truncatulinoides*. The mean standard deviation for all 3 species is around 8%: 7.8%
for *G. bulloides*, 6.4% for *N. incompta* and 13% for *G. truncatulinoides*. We take
these values as the error adjustment for SBW in the different size fractions (250-300
$\mu$m, 200-250 $\mu$m and 400-500 $\mu$m respectively). It is difficult to compare these results
with other studies as size fractions and species are often different, but this error
estimates are in the same order of magnitude as some other MBW published in core-
tops records and sediment traps (de Moel et al., 2009; Moy et al., 2009).
These findings highlight the fact that the use of sieve fractions does not provide
enough control on the influence of morphometric parameters in test weight.
Morphometric variations described in table 1 indicate that the typically used sieve
fractions may be unreliable due to the number of individuals out of the desired
fractions and the variability within the size range. The correlations between SBW
and shell area are consistent with previous studies (Aldridge et al., 2012; Beer et al.,
2010) and underscore the importance of isolating the component of variation in
foraminifera shell thickness that represents a change in calcification and does not
occur as a direct result of change in shell size. Thus, the shell weight was size-
normalized after Beer et al., (2010) by isolating the influence of isometric scaling on
wall thickness and calcification density.
Moreover, both $MBW_{area}$ (Fig. 2) and $MBW_{diam}$, in either the sediment trap data and
core-top data, do not correlate with area and diameter ($1x10^{-5} < r^2 < 0.33$ and 0.001
$< r^2 < 0.2$ respectively) indicating that size does not have an influence on these
values. This suggests that our size-normalization procedure adequately removes the
size influence (Fig. 2) and therefore, our MBW data represents a robust parameter
reflecting test wall thickness and calcification intensity not influenced by test size
(Table 3). Therefore, MBWs can be considered as a reliable calcification intensity
proxy.
Based on all the above, from this point we'll focus our discussion on the $MBW_{area}$ to
discuss the foraminifera shell weight variability on seasonal, interannual and pre to
post-industrial Holocene time scales.

**4.2. Seasonal variations of foraminifera calcification in the NW Mediterranean**

$MBW_{area}$ values were calculated for the three species to illustrate the seasonal variability of these parameters (Fig. 3). Samples have been assigned to their corresponding month according to the mean cup sampling date.

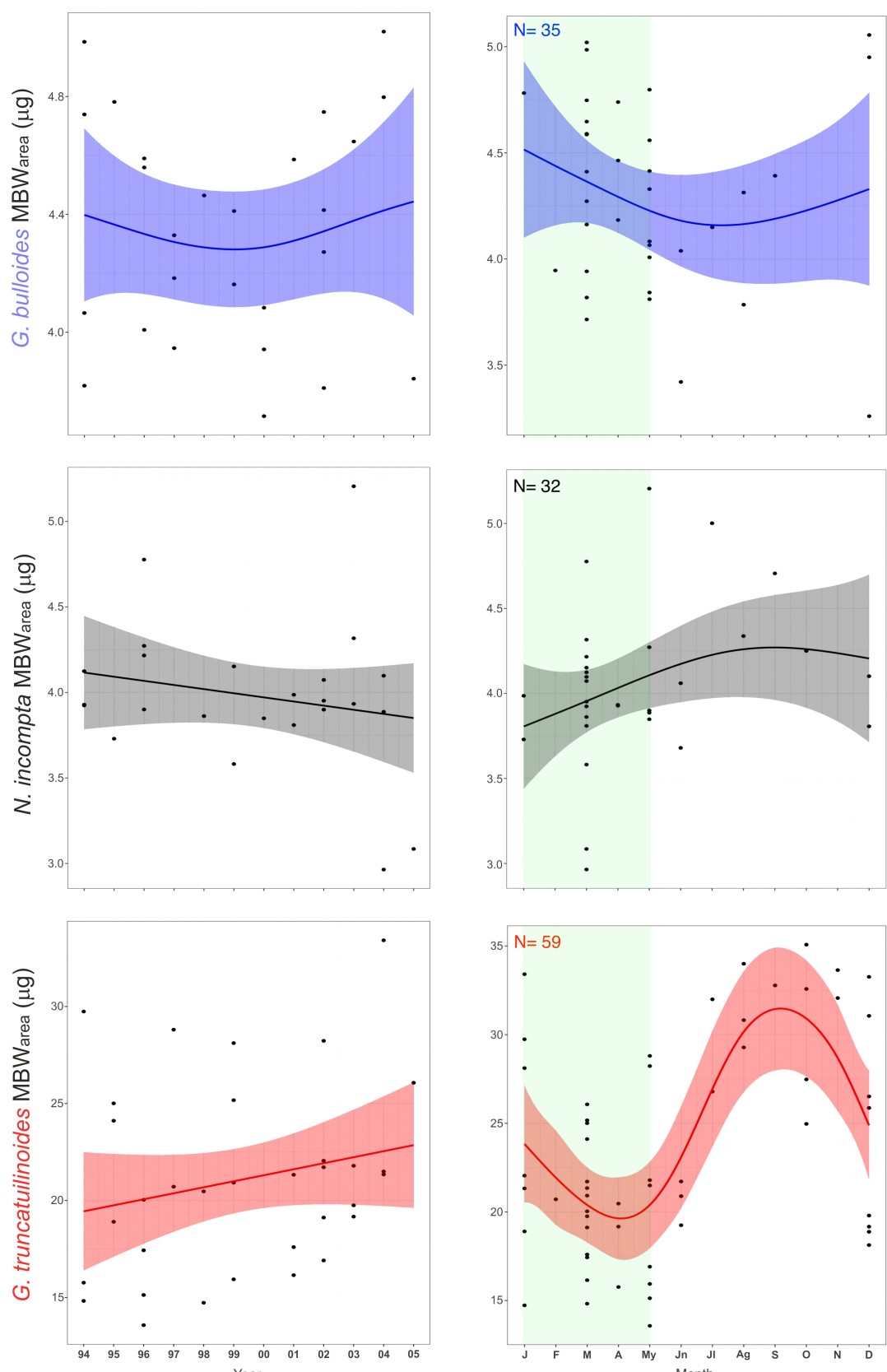

**Figure 3.** MBW$_{area}$ ($\mu$g) values across the years and for a composite year for *G. bulloides, N. incompta* and *G. truncatulinoides* in the Planier sediment trap. Light-green shaded area represents the high productivity period in the study zone (Rigual-Hernández et al., 2012).

The mean MBW$_{area}$ for the three species in the Planier sediment trap are 4.29 $\mu$g ($\pm$ 0.45 $\mu$g for *G. bulloides*, 4.04 $\mu$g ($\pm$ 0.4 $\mu$g) for *N. incompta* and 23.25 $\mu$g ($\pm$ 6.2 $\mu$g) for *G. truncatulinoides*.

The seasonal variations in shell calcification differ according to the species.

In the case of *G. bulloides*, maximum annual calcification values are reached during winter and early spring: 5.05 in December and 5.02 $\mu$g in March. January displays the highest mean value: 4.78 $\mu$g. Minimum values are reached during summer: 3.72 $\mu$g in June, which is also the month that exhibits the lowest mean MBW$_{area}$. Overall, there is a 1 $\mu$g seasonal difference in calcification between maximum and minimum values, which corresponds to a 24.5% change in the mean MBW$_{area}$ value. Mean seasonal standard deviation is $\pm$ 0.47 $\mu$g.

*N. incompta* shows a maximum in calcification in late spring to mid-summer: a maximum value of 5 $\mu$g is reached in May, while July is the month that displays the highest mean value (5 $\mu$g). Lowest values are reached in early spring: 2.96 $\mu$g in March, while January displays the lowest mean value (3.85 $\mu$g). Thus, the annual mean seasonal amplitude is 1.15 $\mu$g which translates into a 28% seasonal MBW$_{area}$ variability. Standard deviation is $\pm$ 0.28 $\mu$g.

Finally, *G. truncatulinoides* displays a seasonal maximum MBW$_{area}$ value in late summer-autumn, with a maximum reached in October: 35.07 $\mu$g, while November; is the month that shows the highest mean MBW$_{area}$ value (32.85 $\mu$g). The lowest value is reached in March: 13.57 $\mu$g, and April is the month that shows the lowest mean value: 18.45 $\mu$g. Seasonal MBW$_{area}$ difference is 14.3 $\mu$g: a 60% variability. Mean typical seasonal deviation is $\pm$ 3.7 $\mu$g.

## 4.3. Interannual MBW$_{area}$ trends

Trends throughout the 12-year record are represented in Fig 3. In order to obtain representative data for each year, maximize data availability of each species and avoid the impact of months with insufficient specimens on the interannual trends, only MBWs from the productive period (January to May) of each year analyzed were included.

*G. bulloides* MBW$_{area}$ showed a slight decrease from 1994 to 2000 and a slight increase from 2000 to 2006. Over the studied interval, the lowest value is reached in the year 2000 and the highest in 2004. Lowest mean annual values were reached during years 2000 and 2005 (3.9 and 3.85 $\mu$g, respectively).

On the other hand, *N. incompta* MBW$_{area}$ showed, a slight calcification reduction with
the highest variability in recent years. Both maximum and minimum values are
displayed in recent years: 2004 and 2005 respectively. Mean yearly MBW$_{area}$ values
reach a maximum in 2003 (4.4 $\mu$g) and a minimum in 2005 (3.2 $\mu$g).
Finally, *G. truncatulinoides* MBW$_{area}$ showed a different pattern, with an overall steep
calcification increase throughout the record. Minimum calcification values are
observed in 1996, which is also the year with the lowest mean MBW$_{area}$ (16.5 $\mu$g)
observed. Maximum value is displayed in 2003, and its mean value is also the
highest of the record (26.1 $\mu$g).
All environmental parameters showed variations across the years. Sea Surface
Temperatures (SSTs) displayed a slight but constant decrease over the years, while
salinity showed a slight increase, mainly since 2002. From late 2000 until late 2002,
phosphate and nitrate concentrations were exceptionally low (Fig. 5). This feature
has already been described in the Gulf of Lions (Meier et al., 2014). Between the 2
periods for which direct *in situ* carbonate system parameters measurements were
available, 1998 to 2000 and 2003 to 2005 (Fig. 5), $CO_3^{2-}$ dropped by 10-15 $\mu$mol/kg,
DIC increased by 40 to 60 $\mu$mol/kg, leading to a pH decrease of 0.02 to 0.025.

**4.4. Sediment trap, core tops and sediment cores MBW patterns**
Foraminifera weights analyzed in core tops and sediment cores from the NW part of
the Mediterranean (Fig. 6) and radiocarbon dating allowed a further insight on
~~reduced~~ foraminifera calcification during the Holocene.
Flux-weighted MBWs (see section 3.4) from Planier sediment trap for the three
planktic species were 4.1 $\mu$g for *G. bulloides*, 3.3 $\mu$g for *N. incompta* and 22.3 $\mu$g for
*G. truncatulinoides* (Fig. 6).
Data from Planier core-top showed higher mean MBW$_{area}$ values: 5.3 $\mu$g, 4.65 $\mu$g
and 35.4 $\mu$g.$^{14}$C dating carried out in this core-top was out of the calibration range
(see section 3.7 for more details), implying that this sample could be considered
post-industrial. Compared to the flux-weighted MBWs from the sediment trap, *G.*
*bulloides* weight has been reduced by 1.2 $\mu$g, *N. incompta* by 1.3 $\mu$g and *G.*
*truncatulinoides* by 12-13 $\mu$g.
Located west of Planier site, Lacaze Duthiers sediment core mean MBWs were:
4.99 $\mu$g for *G. bulloides*, 4.14 $\mu$g for *N. incompta*, and 32.9 $\mu$g for *G. truncatulinoides*.
$^{14}$C analysis displayed a post-industrial age (see section 3.7) for this sample and
corresponding MBWs from this sample for *G. bulloides*, *N. incompta* and *G.*
*truncatulinoides* respectively were: 4.7 $\mu$g, 4.2 $\mu$g and 34 $\mu$g. Overall, compared to
the data from the sediment trap, this corresponds to a 0.6 $\mu$g weight loss for *G.*
*bulloides*, 0.9 $\mu$g for *N. incompta* and 12.2 $\mu$g for *G. truncatulinoides*.
Finally, in the Gulf of Minorca, northwest of Planier site, Minorca sediment core mean
MBWs were: 5.4 $\mu$g for *G. bulloides*, 4.5 $\mu$g for *N. incompta* and 36.3 $\mu$g for *G.*
*truncatulinoides* (Fig. 6). $^{14}$C dating on this core top was carried out on an
intermediate depth (see section 3.7) due to the lack of availability of enough
specimens in the core-top and displayed a date of 1560 calendar years BP (Table
1). Corresponding MBWs for this sample were 5.4 $\mu$g, 4.9 $\mu$g, 38.2 $\mu$g for the three
species. Therefore, the weight reduction compared to the sediment trap flux-
weighted MBWs are: 1.3 $\mu$g for *G. bulloides*, 1.6 $\mu$g for *N. incompta* and finally, 16 $\mu$g
for *G. truncatulinoides.*
**Table 4.** Mann-Whitney variance test results between the MBW$_{area}$ of the
different sites for the three species. Significant values (p<0.05) are set in bold.

| | | PLA ST | PLA CT | LCD SC | MR 3.1.A |
|---|---|---|---|---|---|
| | | | | MBW$_{area}$ | |
| *G. bulloides* | | | | | |
| PLA ST | MBW$_{area}$ | | 0.110 | **0.003** | **7.86e$^{-13}$** |
| PLA CT | | 0.110 | | 1 | 1 |
| LCD SC | | **0.003** | 1 | | 0.114 |
| MIN SC | | **7.86e$^{-14}$** | 1 | 0.114 | |
| *N. incompta* | | | | | |
| PLA ST | MBW$_{area}$ | | 0.438 | 0.890 | **2.59e$^{-5}$** |
| PLA CT | | 0.438 | | 0.342 | 1 |
| LCD SC | | 0.890 | 0.342 | | **0.034** |
| MIN SC | | **2.59e$^{-5}$** | 1 | **0.03** | |
| *G. truncatulinoides* | | | | | |
| PLA ST | MBW$_{area}$ | | 0.120 | **0.003** | **3.13e$^{-15}$** |
| PLA CT | | 0.120 | | 0.644 | 1 |
| LCD SC | | **0.003** | 0.644 | | **0.01316** |
| MIN SC | | **3.13e$^{-15}$** | 1 | **0.013** | |

A Mann-Whitney variance test (see section 3.6) was carried out in order to analyze
the variance between the different MBW$_{area}$ datasets (Table 4) from the different
sites. MBW$_{area}$ data from the sediment trap appeared to have a significantly different
variance compared to the MBW$_{area}$ from Menorca sediment core for the three
species (3.13e$^{-15}$<p<2.59e$^{-5}$), however, differences between the sediment trap data
and the with Lacaze-Duthiers sediment core were only significant for *G. bulloides*
and *G. truncatulinoides* (p=0.003). Concerning differences between the Planier
sediment trap and the underlying core-top, no significant differences were observed
for any of the species (0.11<p<0.438), most likely due to the small number of
samples from the latter site: only 2 samples were available. Note that the differences
between the sediment cores MBW$_{area}$ datasets differ according to the site and
species. In the case of *G. bulloides*, no significant differences are observed between
Planier core-top, Lacaze-Duthiers sediment core and Menorca sediment core. In the
case of *N. incompta* and *G. truncatulinoides*, differences between Lacaze-Duthiers
and Menorca sediment core are significant (0.013<p<0.03), although on lower
orders of magnitude compared to the differences between the sediment trap and
sediment cores datasets (Table 4). This demonstrates that the difference between
the sediment trap MBW$_{area}$ dataset and the seabed sediments MBW$_{area}$ datasets is
greater than the difference between the different seabed MBW$_{area}$ datasets.
## 5. Discussion

**5.1. Seasonal controls on planktic foraminifera shell calcification in the NW**
**Mediterranean**
As described in section 4.2, the seasonal variability of MBW$_{area}$ displays important
differences across the three species analyzed. The different seasonal pattern in
MBW$_{area}$ is reflected by the lack of correlation between the seasonal pattern of
MBW$_{area}$ of the different species, i.e., r= -0.23 (p>0.05) between *G. bulloides* and *N.*
*incompta* and r= 0.16 (p>0.05) between *G. bulloides* and *G. truncatulinoides*. Only
the seasonality of *N. incompta* MBW$_{area}$ and *G. truncatulinoides* MBW$_{area}$ share some
similarities, as reflected in the significant and positive correlation (r= 0.66; p<0.05).
In order to examine the main controls on foraminifera seasonal calcification in the
Gulf of Lions, here we compare the seasonal variability of planktic foraminifera
calcification with foraminifera fluxes previously estimated for the Planier sediment
trap (Rigual-Hernández et al., 2012) satellite data for the studied site and a suite of
environmental parameters measured at the DYFAMED site (see section 3.4).
Furthermore, GAM have been generated for all three species (see Supplementary
figs. 3,4 and 5) and the environmental parameters considered here in order to give
a further insight on the potential factors controlling the MBW$_{area}$. These models
showed that *G. bulloides* and *G. truncatulinoides* seasonal calcification trends are
significant (p=0.05 and p=2.4e-5 respectively). On the other hand, *N. incompta*
seasonal trend does not appear to be significant (p=0.14).
**Table 5**. Correlation matrix of seasonal (monthly) test weights and the
environmental parameters from Planier (sediment trap and satellite data) and
DYFAMED site (see section 3.4). Significant correlations (p<0.05) are set in
bold.

| Parameters | | G. bull. MBW$_{area}$ | N. inc. MBW$_{area}$ | G. truncat. MBW$_{area}$ | G. bull. Fluxes | N. inc. Fluxes | G. truncat. Fluxes | Chl-a | SST | Salinity | [NO$_3$] | [PO$_4$] | pH | [CO$_3$] | [CO$_2$] |
|---|---|---|---|---|---|---|---|---|---|---|---|---|---|---|---|
| | | **Planier site data** | | | | | | | | **DYFAMED site data** | | | | | |
| G. bull. | MBW$_{area}$ | 1 | 0.232 | 0.167 | 0.012 | 0.027 | 0.152 | 0.318 | -0.32 | -0.163 | 0.292 | 0.33 | 0.096 | 0.189 | 0.243 |
| N. inc. | | -0.232 | 1 | **0.667** | **-0.582** | -0.407 | -0.405 | -0.484 | **0.688** | 0.368 | 0.272 | 0.235 | -0.35 | 0.474 | -0.28 |
| G. truncat. | | 0.167 | **0.667** | 1 | **-0.905** | **-0.725** | **-0.666** | **-0.585** | **0.672** | -0.299 | 0.258 | 0.512 | 0.113 | **0.732** | 0.541 |

Here, we first approach seasonal shell calcification by considering the Optimum Growth Conditions (OGC). Previous studied have defined these conditions on a wide variety of ways: abundance of foraminifera, the chlorophyll-a concentration and even nutrients concentration (de Villiers, 2004; Schiebel et al., 2001; Schiebel and Hemleben, 2017). Therefore, we aim to explore the impact of these parameters as OGC on the shell calcification.

Among all the environmental parameters, de Villiers; (2004) suggested that shell calcification, and therefore MBWs, is primarily controlled by the OGC that can be defined as the most suitable environmental conditions for the development of a given planktic foraminifera species. Based on the latter study, it could be expected that favorable environmental conditions for foraminifera growth would lead to both greater shell fluxes and enhanced shell calcification. Our correlation analysis shows different relationships between foraminifera abundance (as inferred from foraminifera shell fluxes) and shell weight. In particular, *G. bulloides* MBW$_{area}$ shows no correlation with its fluxes (Table 5), while both *N. incompta* and *G. truncatulinoides* exhibit a negative correlation between MBW$_{area}$ and their species fluxes although only significant for the latter species (r= -0.4, p>0.05 and -0.66, p<0.05 respectively). GAM results (see Supplementary figs. 3, 4 and 5) support these observations, with shell flux showing a stronger effect on the calcification for *G. truncatulinoides* compared to the other two species fluxes.

According to the OGC theory, species calcification patterns vary according to the species and their fluxes. Interestingly, *G. truncatulinoides* calcification correlates negatively and significantly with all three species fluxes, a pattern opposite to what the OGC theory predicts (de Villiers, 2004), i.e., optimum ecological niche is associated with enhanced calcification. Thus, a possible explanation reconciling our observations with the OGC theory may be that *G. truncatulinoides* tend to prioritize energy allocation toward growth and reproduction at the price of a reduced calcification. As described previously (see section 3.3), this species life cycle is complex as it migrates through different depth levels of the water column throughout the year. It is thought to reproduce once a year in winter in subtropical waters and it

has been speculated that nutrient availability and the lack of predation could explain
this strategy. Therefore, the observation of peak calcification in autumn/winter (Figs.
3 and 4) may reflect the migration of adults - that have spent time in deeper waters
developing a thick calcite crust - to shallower waters in late autumn to winter to
undergo reproduction. During this interval, the other major species display low
abundances in the water column, which could allow *G. truncatulinoides* to reproduce
due to the lack of competition. *N. incompta* calcification displays a similar pattern, a
negative correlation with all three species, but with a lower level of significance. It's
$MBW_{area}$ correlates negatively and significantly ($p<0.05$) with *G. bulloides* flux, but
its fluxes correlate positively and significantly with the latter species fluxes (see
Supplementary table 2). This is interesting, as it may highlight interspecific relations.
First, this could lead to the assumption that when *G. bulloides* dominates the
assemblages, *N. incompta* also displays a high abundance (Rigual-Hernández et
al., 2012). Then, it could show that when conditions are favorable, *N. incompta*
reproduces at a higher rate at the price of thinner shells (Table 1). This agrees with
*N. incompta* life cycle, which is known to be outnumbered by opportunistic species
when nutrients supply is high (Schiebel et al., 2002), but dominate the assemblages
when stratified waters are set,  therefore when conditions are favorable or when in
cohabitation with opportunistic species, it could focus on its reproduction. Note that
*G. truncatulinoides* and *N. incompta* $MBW_{area}$ correlate positively and significantly
($p<0.05$), showing a similar calcification pattern on a seasonal scale.
An alternative proxy for OGC that may be considered is chlorophyll-*a* concentration.
Chlorophyll is considered an indicator of the algal biomass concentration, which is
known to represent a large part of some foraminifera species diet, specially for *G.*
*bulloides* (Schiebel and Hemleben, 2017). Our data shows a positive but not
significant correlation between chlorophyll concentration and *G. bulloides* MBW (r=
0.32, p>0.05), suggesting a negligible influence of phytoplankton concentration on
*G. bulloides* calcification. A stronger trend would be expected under the OGC theory
as algae are a vital part of its diet (Hemleben et al., 1989). This lack of correlation
between *G. bulloides* and chlorophyll-*a* has already been described (Weinkauf et al.,
2016). We speculate that *G. bulloides* may preferentially feed on certain groups of
phytoplankton which changes in seasonal abundance in the photic zone do not
necessarily follow the seasonal pattern of total chlorophyll concentration (Marty et
al., 2002). In the case of *N. incompta* and *G. truncatulinoides*, correlations between
$MBW_{area}$ and chlorophyll-*a* are negative and stronger although only significant for the
latter species (r= -0.58, p<0.05). GAM results further support these observations
(see Supplementary fig. 5), with chlorophyll-a showing a significant impact on *G.*
*truncatulinoides* calcification. This observation indicates that optimum calcification
conditions for *G. truncatulinoides* are reached at times of minimum annual algal
biomass concentration in the photic zone. It is possible that, due to its deeper habitat
(Schiebel and Hemleben, 2017), *G. truncatulinoides* feeds on phytoplankton
dwelling in subsurface levels of the water column. In fact, a deep chlorophyll
maximum is known to develop during large part of the year in the Northwestern
Mediterranean (Estrada et al., 1993) but its presence is not detected by satellites.
This interpretation is in agreement with earlier work by Pujol and Vergnaud Grazzini
(1995) who found peak abundances of this species during the summer below the
thermocline.

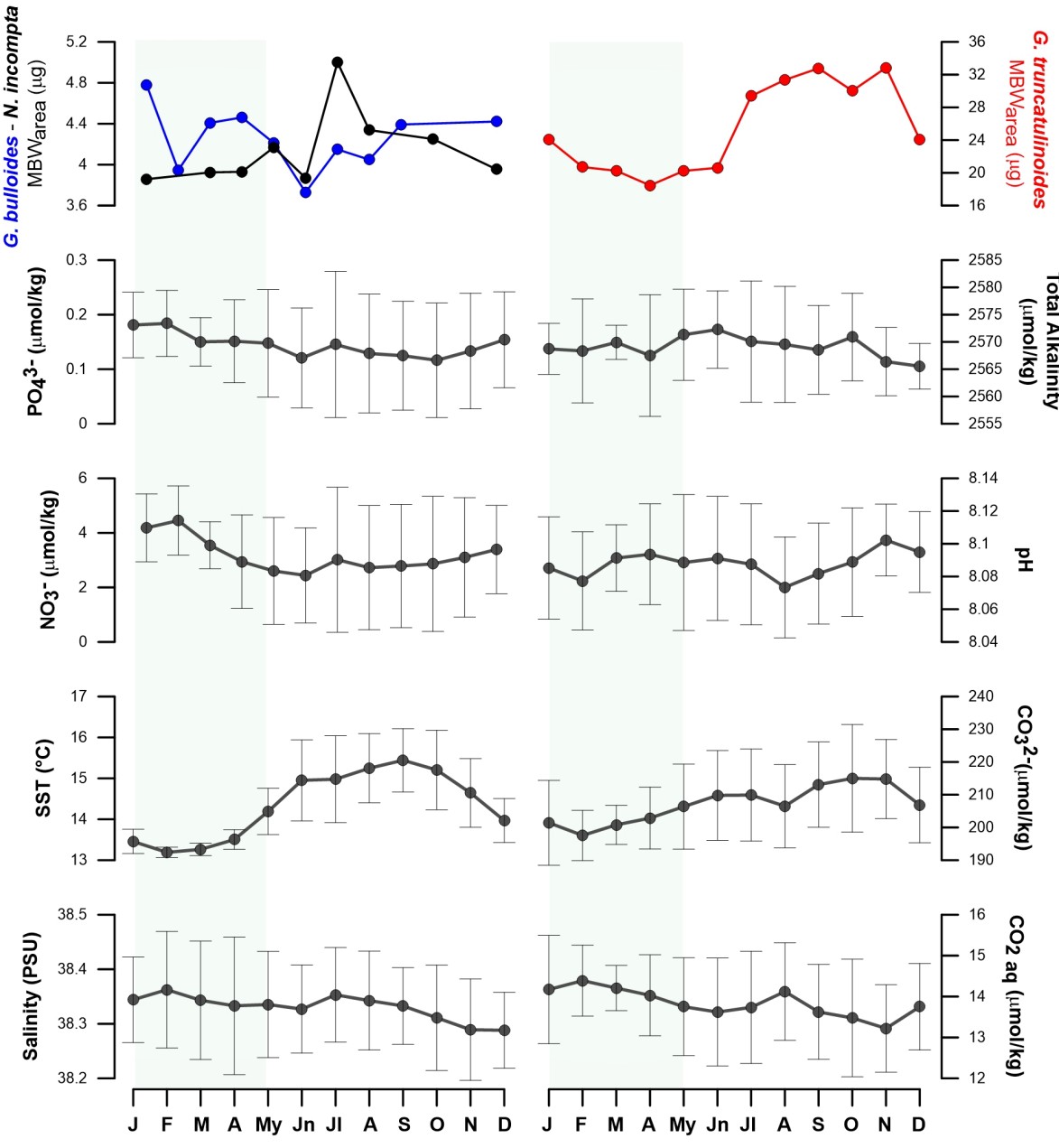

**Figure 4.** *G. bulloides*, *N. incompta* and *G. truncatulinoides* seasonal mean
monthly MBW$_{area}$ variations compared with Planier environmental data and

the resampled seasonal signal of environmental parameters from the
DYFAMED site across a composite year.

Something to consider when using chlorophyll as an OGC proxy is that it can be
confounded with the nutrient concentrations. Previous studies have described that,
in those settings where foraminifera abundance covaries with nutrient
concentrations, then nutrients are probably a better OGC proxy than chlorophyll
concentrations (Schiebel et al., 2001). In turn, the correlation of nutrients (nitrates
and phosphates) with fluxes were positive for all three species, although only
significative ($p<0.05$) for *G. truncatulinoides* abundance (r= 0.58 and 0.59 for nitrates
and phosphates respectively). Although here we have first described the OGC as
species fluxes and then as the chlorophyll-*a* concentration, it is important to
remember that the niche and favorable conditions meant to be described by the OGC
for each species are multi-dimensional.
Note that nitrate and phosphate concentration variations were closely linked to each
other (r= 0.876, $p<0.05$) , making it difficult to determine if the resulting effect on
foraminifera calcification is due to the effect of a single driver or to the combination
of both. High phosphate concentrations are generally considered an inhibitor for
foraminifera calcification (Zeppenfeld, 2019). Evidences show that calcite formation
is inhibited by $PO_4^{3-}$ due to its adsorption on the calcite surface, impeding its
precipitation by blocking the crystal growth. Aldridge et al., (2012) showed a negative
effect of phosphate concentration on *G. bulloides* with $PO_4^{3-}$ concentrations of 0.1-
0.4 $\mu M$. This feature has been shown in other calcareous organisms such as
coccolithophores (Paasche and Brubank, 1994) and calcifying algae (Demes et al.,
2009), resulting in a higher growth and calcification under controlled $PO_4^{3-}$. On the
other hand, no studies show a reduced calcification in marine organisms under high
$NO_3^-$ values (Fig. 4). Our work shows that nutrient concentrations (both nitrates and
phosphates) do not correlate significantly with any of the three species MBW studied,
and this observation is supported by the GAM results which do not show any
significant effect of nutrients concentrations on the calcification. $NO_3^-$ correlations
were not significant for all three species, while $PO_4^{3-}$ correlations were also very
weak, with the exception of the one with *G. truncatulinoides* MBW$_{area}$ (r= -0.512, p=
0.07) (Table 5).
Previous studies have suggested that salinity may have an influence on foraminifera
calcification. Zarkogiannis et al., (2022) found that salinity may control calcification
of certain foraminifera species in the central Atlantic region. However, our data
suggest that the role of salinity on calcification in our study region is unlikely since
its seasonal amplitude is tiny (0.1 PSU; Fig. 4). This idea is supported by the lack of
correlation between salinity and MBW$_{area}$ for the three species studied (Table 5) and
the GAM results, in which salinity was the parameter that displayed the lowest effect
on calcification.
Temperature (Sea Surface Temperature) has been described as a major factor that
controls the size (Schmidt et al., 2004) and porosity (Burke et al., 2018) of planktic
foraminifera, therefore it could represent a major control factor on shell calcification
in the NW Mediterranean. In particular, calcification could be positively linked to
temperature through different mechanisms: (i) warmer temperatures have been
shown to increase enzymatic activity and therefore enhanced growth and
calcification rates (Spero et al., 1991); (ii) Lombard et al., (2011) stated that higher
temperatures could also increase feeding and ingestion rates, but it remains unclear
if this could result in a calcification rate increase. Our data revealed that SST
correlates negatively, but without significance with *G. bulloides* calcification, while
correlations between SST and *N. incompta* and *G. truncatulinoides* were both
positive and significant (r= 0.69 and 0.67 respectively, $p<0.05$). GAM results also
displayed a positive and the most significative effect of the SST on these two
species. These findings highlight that SSTs are one of the main factors affecting *N.*
*incompta* and *G. truncatulinoides* calcification among the parameters considered
here. Finally, in addition to having an impact on the size and calcification of the
planktic foraminifera, temperature is well known as a major control of the carbonate
system, due to the increased solubility of atmospheric $CO_2$ at lower temperatures,
and therefore it could have an indirect effect on foraminifera calcification by affecting
the carbonate system.
Data for the carbonate system were only available for years 1998 to 2000 and 2003
to 2005 and, therefore gaps comprised in these years were filled with estimates
using the CO2sys macro (see section 3.6 for more details). However, note that the
data available for these parameters was relatively smaller compared to the other
parameters and may have prevented detection of other significant relationships. The
relationship between $CO_3^{2-}$ and MBW has been described in previous studies
(Barker and Elderfield, 2002; Marshall et al., 2013) and the bulk of evidence indicates
that foraminifera MBWs to be positively linked with $CO_3^{2-}$ concentrations (Aldridge et
al., 2012; Osborne et al., 2016). However, it appeared that planktic foraminifera
response to $CO_3^{2-}$ concentration was not uniform and varied across species (Beer
et al., 2010; Lombard et al., 2010). The trends between carbonate system
parameters and MBWs were similar to those observed when comparing MBWs with
temperature, highlighting the covariations between these two parameters (Fig. 4).
Our data showed that $CO_3^{2-}$ concentrations were not significantly correlated with *G.*
*bulloides* MBW$_{area}$ nor with *N. incompta*. Only *G. truncatulinoides* MBW$_{area}$ displayed
a clear significant correlation with $CO_3^{2-}$ concentration (r= 0.73, $p<0.05$), implying
that carbonate availability may represent a key control on *G. truncatulinoides* in the
Northwestern Mediterranean. On the other hand, $CO_2$ concentrations, excepting a
negative and almost significant relationship with *G. truncatulinoides* $MBW_{area}$ (r= -
0.54, p= 0.06), showed no correlation with the other 2 species MBWs. None of the
remaining carbonate system parameters (pH, alkalinity and calcite saturation),
exhibited a significant seasonal correlation with the MBWs. On the other hand, GAM
result (see Supplementary figs. 3, 4 and 5) did not show a significant impact of any
carbonate system parameters for any of the three species calcification. As stated
previously, the lack of data could have prevented the detection of further trends, but
considering the seasonal patterns of carbonate system parameters, a potential role
of the $CO_3^{2-}$ concentration could be expected.
In summary, seasonal correlations, trends and GAM showed that the environmental
parameters that displayed the highest correlation with $MBW_{area}$ vary according to the
species. *G. bulloides* calcification appeared to be affected mainly by interspecific
relations. *N. incompta* calcification showed to be mainly positively linked to the SST.
Finally, *G. truncatulinoides* calcification was positively linked with the SST and
potentially $CO_3^{2-}$ concentration, while OGC displayed a negative effect on its
$MBW_{area}$. The combined effect of these parameters seems to control foraminifera
calcification in the Gulf of Lions; however, it should be considered that covariation
between these parameters is strong, and therefore it is difficult to isolate the effect
of a single parameter. Moreover, it is likely that the ecology and life cycle of the
species could also be a major factor affecting the response of the species
calcification to the environmental parameters variations. Our results are in
agreement with earlier studies that stated that OGC (de Villiers, 2004), temperatures
and $CO_3^{2-}$ (de Villiers, 2004; Marshall et al., 2013; Osborne et al., 2016)
concentrations are the main factors that impact calcification in planktic foraminifera,
while the calcification response to those parameters is species-specific, which is in
agreement with the work of Weinkauf et al, (2016).

**5.2. Interannual trends in planktic foraminifera calcification**
As stated previously, the Mediterranean Sea is a sensitive zone to atmospheric $CO_2$
accumulation (Ziveri, 2012) and is experiencing ongoing ocean acidification. On an
interannual time scale, different studies (Beer et al., 2010; Osborne et al., 2016)
have shown that sea surface warming and carbonate system parameters are the
most likely parameters to control calcification on key calcifying phytoplankton
species such as the coccolithophore *Emiliania huxleyi* organisms (Meier et al.,
2014). However, datasets from sediment traps that cover a wide span of years and
in which foraminifera weights have been analyzed are rare (Kiss et al., 2021),
therefore it is difficult to place our results in a more global context. Our GAM results
(see Supplementary figs. 3 and 4) showed that both *G. bulloides* and *N. incompta*
interannual patterns were non-significant. This is not surprising as the calcification
trends for these two species did not display a clear and marked variation over the
years, excepting a small mean calcification reduction (Fig. 3) and minimum
calcification values in 2004 and 2005 (Fig. 3 and Supplementary figs. 3 and 4).

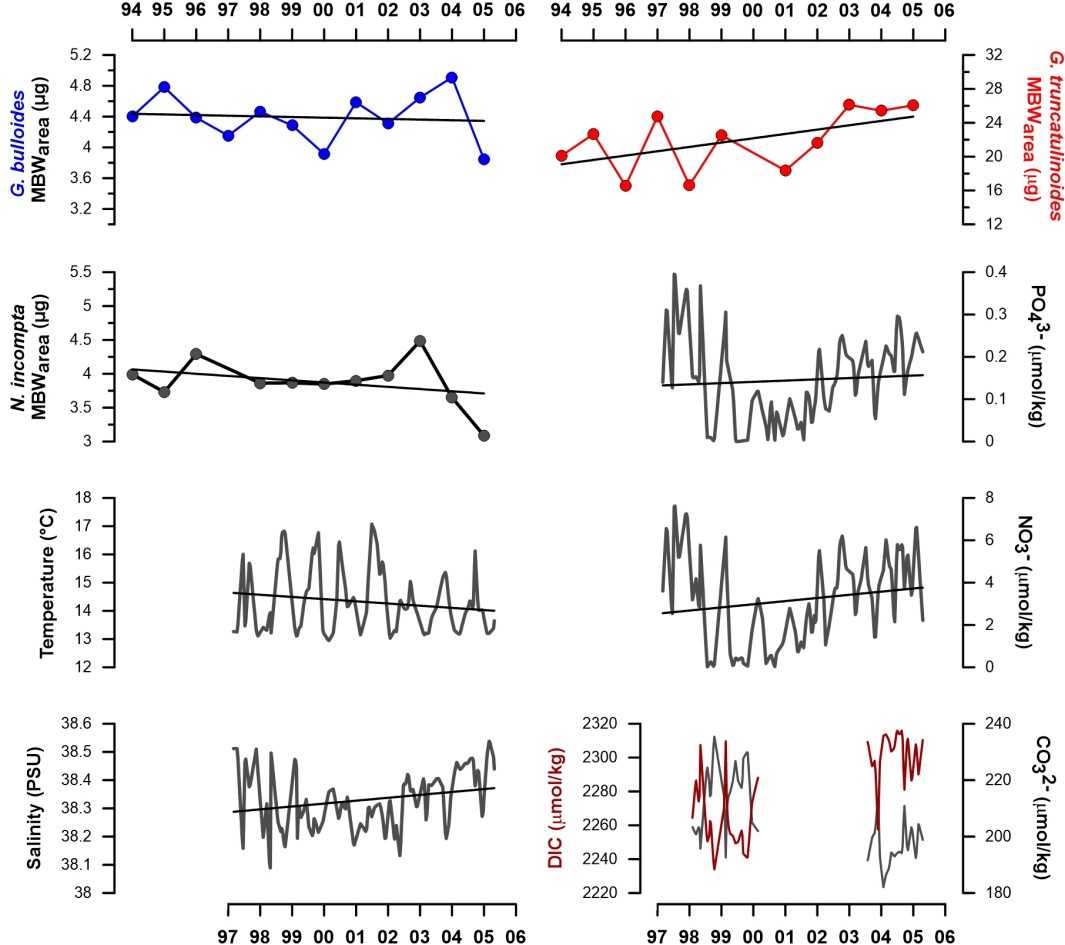

**Figure 5.** Interannual mean MBW$_{area}$ ($\mu$g) values for *G. bulloides*, *N. incompta*
and *G. truncatulinoides* from the high productivity period (see section 2) and
Planier and DYFAMED environmental data variations across the record.
Black lines represent the trends from the MBW$_{area}$ and resampled data. DIC
represents "Dissolved Inorganic Carbon".
Notably, the trend in *G. truncatulinoides* is opposed to the previous two species and
shows a steady and steep increase throughout our record. Over the analyzed time
span, its MBW increased around 20% (equivalent to an increase of ~5 $\mu$g).
According to the GAM results, the interannual calcification trend for this species is
significant (see Supplementary figure 5). If this calcification increase continues on
current trends, then the average MBW of *G. truncatulinoides* will double by 2024.
Analysis of present *G. truncatulinoides* populations is urgently needed to assess if
the observed trend held true during the last two decades. It is important to note that
while *G. truncatulinoides* exhibits an intimate correlation with $CO_3^{2-}$ concentration on
a seasonal scale, no clear correlation was found with the interannual changes of
$CO_3^{2-}$ concentration. This feature is also supported by the GAM results. A similar
enhancement in shell calcification has been described in the Balearic Sea for *G.*
*truncatulinoides* in high-resolution sediment cores (Pallacks et al., 2020), but also in
*Globorotalia inflata*. Taken together, our observations and the study mentioned
above, suggest that deep dwellers are unaffected by the recent ocean acidification
and changes in the carbonate system and that the recent change in one or several
environmental drivers may be stimulating the calcification of these species.
Here, we theorize that the interannual patterns presented in Figs. 3 and 5 mainly
reflect the seasonal changes in the regional oceanographic setting. As described
previously (see section 2 for more details), the Gulf of Lions is influenced by a strong
seasonality. The recent SST decrease could be linked to an enhancement in water
mixing, as cold and deep salty water reach up to the surface. This mechanism would
be less intense during years 2000 to 2002, corresponding to a SST increase along
with a salinity decrease and absolute minimums in nutrients concentrations (Fig. 5),
as water stratificates, these are consumed by primary production. Finally, in recent
years, water mixing seems to be reactivated, as SST keeps decreasing and nutrients
concentrations increase again. This mechanism also affects the carbonate system
parameters, as water mixing brings to surface deeper DIC enriched waters to the
surface, coupled with a $[CO_3^{2-}]$ reduction. Our data shows that alkalinity patterns
display similar tendencies to DIC, however, until the second time span covered by
carbonate system data, alkalinity variations are proportionally higher than DIC
variations (see Supplementary material), suggesting a water mixing phenomenon.
On the other hand, DIC variations turn to be higher than alkalinity variations from
2003 to 2005, suggesting an additional effect of carbon inputs on the carbonate
system not reflected in the alkalinity data.
Note that SSTs, despite showing a positive and significant correlation with *N.*
*incompta* and *G. truncatulinoides* on a seasonal scale and the GAM showing a
positive and significant effect on the calcification, did not follow the same pattern as
the latter species. This observation implies that other mechanisms or parameters
than the ones considered here may be affecting the $MBW_{area}$ on an interannual
scale.
**5.3. Holocene core-top data comparison**

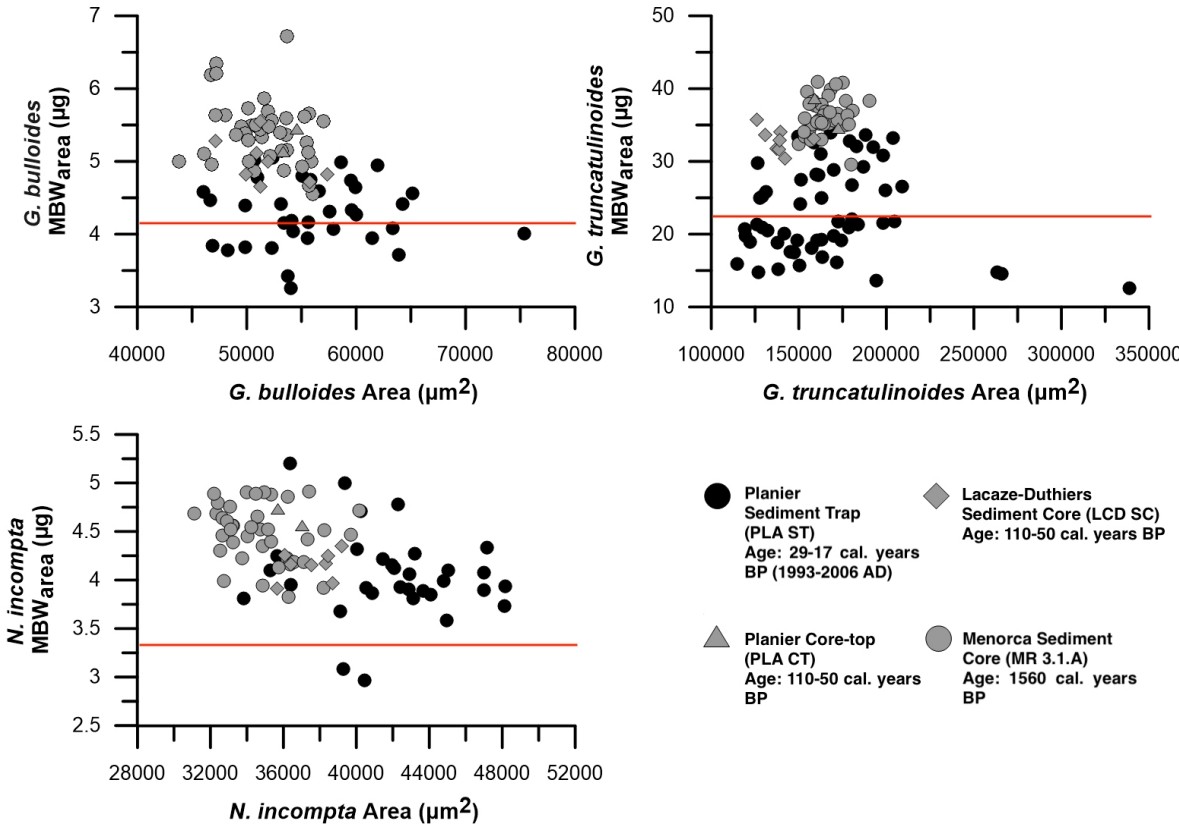

**Figure 6.** MBW$_{area}$ in μg and area in μm$^2$ comparison in the sediment trap (PLA ST), Planier core-top (PLA CT), and both Lacaze-Duthiers (LCD SC) and Minorca sediment core (MR 3.1.A). Black dots represent data from the sediment trap, while lighter colors represent data from the different seabed sediments. Red lines represent the flux-weighted values from the sediment trap.

The comparison of the well-preserved assemblages of planktic foraminifera in the pre-industrial and industrial Holocene-aged surface sediments with those collected by a long-sediment trap record offers a unique opportunity to assess the impact of recent environmental change on the calcification of calcareous zooplankton in the Mediterranean Sea (Fig. 6). However, when comparing data from sediment traps and seabed sediments, the possible role of calcite dissolution must be taken into account.

Calcite dissolution in the water column and/or on the sea floor could be invoked as a source of variability between the sediment trap and surface sediment data sets (e.g., Dittert et al., 1999). Therefore, in order to obtain meaningful interpretations from our data sets it is important to assess the possible role of dissolution in the preservation of planktic foraminifera shells. Several lines of evidence suggest, however, that calcite preservation does not represent an important source of bias in our study area. Firstly, the Mediterranean Sea is supersaturated with respect to calcite (Millero et al., 1979) and the location of all the analyzed samples is much

shallower than the location of the calcite saturation horizon (Álvarez et al., 2014),
therefore, calcite dissolution seems unlikely (Schneider et al., 2007). Secondly,
several sediment trap studies have documented that calcareous plankton
experience negligible dissolution in their transit from the surface ocean to the sea
floor (Beaufort et al., 2007; Moy et al., 2009; Rigual-Hernández et al., 2020). Thirdly,
SEM and microscopic observations of all 3 species in samples from both the
sediment traps and sediment cores showed no sign of dissolution and foraminifera
were well preserved (see Supplementary fig. 6). These arguments suggest that
calcite dissolution does not represent an important control in the weight of the
planktic foraminifer shells in the analyzed samples. However, it has been
documented that when dissolution takes place, the thinnest shells are affected first
(Berger, 1970) while the heaviest and more calcified specimens remain. In our study,
the specimens from the sediment trap were lighter that the ones from the sediment
cores, therefore, this is important to acknowledge as dissolution cannot be
completely ruled out as a possible source of variability between the surface sediment
and sediment trap data sets.
Overall, the lower shell weights of the foraminifera collected by the traps strongly
suggest that the three planktic foraminifera species have experienced a reduction in
their calcification since the industrial era and/or late to recent Holocene. While the
shell weight of each species measured in the sediments show some variability
across seabed sediments (Fig. 6), our data suggest an overall reduction of 18-24%
for *G. bulloides*, 20-27% for *N. incompta*, and 32-40% for *G. truncatulinoides.* It is
important to note that the range of shell weight variability across core-tops and
sediment cores (4.5-6.7 $\mu$g and 0.37 $\mu$g typical deviation for *G. bulloides*, 3.8-4.9 $\mu$g
and 0.23 $\mu$g typical deviation for *N.incompta*, and 29.5-40.9 $\mu$g and 2.6 $\mu$g typical
deviation for *G. truncatulinoides*) is substantially lower than the difference with the
sediment trap data (3-5 $\mu$g and 0.5 $\mu$g typical deviation for *G. bulloides*, 2.9-5.2 $\mu$g
and 0.5 $\mu$g typical deviation for *N. incompta* and 12-35 $\mu$g and 6 $\mu$g typical deviation
for *G. truncatulinoides*), implying that the shell weight of recent foraminifera
populations for the three species is lower than anywhere in the NW Mediterranean
in the pre-industrial and post-industrial times. The source of the variability across
core tops and sediment cores is most likely caused by the different age of the
samples, ranging from 1560 cal. years BP at Minorca mid-depth (Table 1) sample to
post-industrial at Planier and Lacaze-Duthiers core-tops, and the different
environments associated to the location of each core top.
A non-parametric two-way Mann-Whitney test (see sections 3.6 and 4.3) showed
that the sediment trap MBW$_{area}$ dataset was significantly different ($p<0.05$) from MR
3.1.A and non-different from PLA CT for all three species studied here (Table 5).
Something to consider when comparing recent sediment trap data with pre-industrial
Holocene data is the life cycle of the species. As all the species analyzed presented
some reduction in shell calcification, the degree to which the different specimens
responded varied. The greatest weight reductions were observed for *G.*
*truncatulinoides* populations, while *G. bulloides* populations exhibited the lowest
weight loss.
Previous work stated that those species hosting photosynthetic algal symbionts
exhibit a higher tolerance to environmental changes that may affect their calcification
(Lombard et al., 2009). This is due to the fact that these symbionts can modify the
sea water chemistry that is close range to the shell, allowing a calcification
enhancement. Of the species studied here, none are known to be symbiont bearing
species, with the possible exception of *G. bulloides*, therefore, they are among the
most vulnerable foraminifera species to any sea water chemistry change.
It has been described that some morphotypes of *G. bulloides* do actually host
bacterial endobionts in their cytoplasm (Bird et al., 2017). The later work showed
that high amounts of *Synechococcus*, a cyanobacteria, were found in morphotype
IId specimens *of G. bulloides* from the California coast. Although no such
observations have been reported on morphotype Ib, the dominant *G. bulloides*
morphotype in the Mediterranean sea (Schiebel and Hemleben, 2017), this could be
relevant as bacterial photosynthetic activity would interact on the close range
seawater chemistry by removing $^{12}CO_2$ and therefore impacting the $^{13}C/^{12}C$ ratios in
the surrounding dissolved $CO_2$. Moy et al., (2009) work in the Southern Ocean,
showed a 30-35% calcification reduction for *G. bulloides* during the industrial era.
Our study shows that such a similar reduction in *G. bulloides* $MBW_{area}$ (i.e., a mean
20% taking into account the 3 sites studied) has also taken placed in the
Mediterranean Sea.
Even though the species studied were different in Fox et al., (2020), and that shell
thickness was analyzed, that work showed a massive shell reduction for *N. dutertrei*
(around 75%) and a smaller reduction for *G. ruber* (around 20%). *N. incompta* weight
reduction in this study is around 25%, despite that life cycles are different between
these species, our results come in the same line.
Data for *G. truncatulinoides* calcification comparison between pre-industrial
Holocene and post-industrial Holocene is scarce. One of the few available studies is
the one of Pallacks et al., (2020) in the western Mediterranean sea using pre-
industrial data and recent foraminifera weight data obtained from high resolution
core-tops. Size-normalized weights showed that all the species calcification
decreased since the onset of the industrial revolution, an observation that is
supported by our data (Fig. 6). *G. truncatulinoides* showed a 24% weight reduction,
which is a lower reduction than what is shown in our study (around 35% $MBW_{area}$
decrease), but shows a similar trend. Taken together, all these observations suggest
that a decrease in major planktic foraminifera calcification is not only a regional
feature but a global scale process.
On a more regional scale, Hassoun et al., (2015) documented the ongoing changes
in seawater carbonate speciation in the Mediterranean waters. In the latter work, the
distributions of anthropogenic $CO_2$ showed that all Mediterranean water masses
have already experienced ocean acidification. This effect was more pronounced in
the intermediate to deep masses (300-500m and >500m respectively) in the western
basin, which translated into a minimum pH reduction of 0.1 in this part of the
Mediterranean. As stated previously, over the years in which carbonate parameters
were retrieved from the DYFAMED database, pH was reduced, DIC showed a
marked increase and $[CO_3^{2-}]$ displayed a decrease. Taken together these
observations and our data, it is possible that the observed changes in foraminfera
calcification could have been totally or partially driven by the ongoing ocean
acidification in the Mediterranean.
Moreover, the largest calcification reduction is observed between the seabed
sediments and the sediment trap, this means that the highest calcification reduction
has taken place between post-industrial Holocene and recent Holocene (i.e. the
reduction between LCD SC/PLA CT and PLA ST) (Fig. 6). This could be explained
with the "Great Acceleration theory". The Great Acceleration is a term used to
describe the trends in $CO_2$ emissions and the associated temperature changes as
consequences of the human impacts on the atmosphere since the 1950s (Head et
al., 2022a, 2022b).
However, other important changes in physical and chemical parameters co-occur
with ocean acidification, and therefore should be also considered. Based on the
seasonal and interannual patterns of the SST in the Gulf of Lions (Figs. 4 and 5),
temperature trends could also be invoked as a likely parameter to affect calcification
here. As shown by the correlations (Table 5) and the GAM results, SSTs are one of
the most likely parameters to affect calcification on different timescales. However,
on a pre-industrial to post-industrial timescale, the effect of the SST on the
foraminifera calcification may be hard to evaluate due to the effect of the latter on
the carbonate system parameters such as $CO_2$ and $CO_3^{2-}$ concentration in the water
But note that the Mediterranean is considered to be warming at a faster rate than
the global average (Hassoun et al., 2015; Lazzari et al., 2014).  Calcification data
from the sediment trap has been flux-weighted (see section 3.4) in order to be
compared with the sedimentary calcification data, therefore, this data could be
affected by a change in the incoming foraminifera flux (de Moel et al., 2009). In this
line, the Gulf of Lions, presents a marked seasonality (see section 2) and the both
the mass fluxes (Heussner et al., 2006) and foraminifera fluxes present strong
seasonal variations. Parameters such as the North Atlantic Oscillation index, the
river runoff and the intensity of the seasonal water cascading process have been
suggested to play a role on planktic foraminifera production and export (Rigual-
Hernández et al., 2012). The later study shows that most of the species flux showed
a yearly uni-modal distribution, but the flux values and distribution remained fairly
constant over the years. This highlights that, in our study zone, a major change in
the foraminifera flux affecting the flux-weighted calcification value is unlikely.
**5.4. Influence of environmental variability on MBW$_{area}$ across different time**
**scales**
Our results show that the influence of environmental parameters over the different
time scales studied is not constant and depends on the species, the environmental
driver and timescale.
In the case of *G. bulloides*, our data suggest that OGC and inter-specific
relationships seem to affect its MBW on a seasonal scale, carbonate system seems
to play a major role while on an interannual and on a pre/post-industrial time scales.
*N. incompta* calcification seems affected by OGC, inter-specific relationships and
SST on a seasonal scale, while on longer time-scales carbonate system appears to
play preponderant role. Finally, *G. truncatulinoides* calcification seems positively
correlated to carbonate system and SSTs and negatively with the OGC on a
seasonal scale. However, these patterns seem to have an opposite effect on an
interannual scale, as *G. truncatulinoides* calcification shows a clear increase while
carbonate system parameters become less and less favorable for calcification. In
turn, on a pre/post-industrial Holocene time scale, its MBW$_{area}$ seem to be affected
by regional processes such as OA and warming.
Interestingly, this may suggest that other parameters that the ones considered in this
work could also influence planktic foraminifera calcification on different time scales.
Factors such as changes in the regional oceanographic processes (Cisneros et al.,
2019; Durrieu de Madron et al., 2017) affect the physical and chemical properties of
the water column and hence, could impact the life cycle of the species studied here.
Also, while *G. bulloides* can either present regular or encrusted forms; *N. incompta*
and *G. truncatulinoides* are crust forming species. In our study, *G. bulloides*
individuals are mainly regular forms, but encrusted individuals were identified in both
the sediment trap and seabed sediments. It is out of the scope of this work to focus
on the effect of the crust on the species MBW, however, Osborne et al., (2016) study
showed that encrusted *G. bulloides* individuals are around 20-30% heavier than the
regular ones. Finally, reproductive strategy could be invoked as a factor affecting
calcification. Planktic foraminifera mainly reproduce by gametogenesis, however,
asexual reproduction has been documented to happen on a rare but constant rate

(Meilland et al., 2021; Takagi et al., 2020). It has been theorized with overwintering the polar species *N. pachyderma* that if all of the individuals reproduce with an asexual strategy, they would double their size populations in 2 generations, with observations reporting the growth of extra chambers (Meilland et al., 2022). However, it remains to be shown if this phenomenon only affects polar species in such proportions and if other species show similar patterns.

## 6. Conclusions

The variability in shell calcification of three planktic foraminifera species (*G. bulloides*, *N. incompta* and *G. truncatulinoides*) was studied in the northwestern Mediterranean Sea at different time scales using sediment trap and seabed samples. The analysis of 273 samples and more than 4000 individuals revealed that:

i.   The Sieve Based Weight (SBW) method is not a reliable tool as calcification indicator due to the influence of morphometric parameters on foraminifera weight. The Measured Based Weight (MBW) technique, on the other hand, shows little to negligible influence of the morphometric parameters, and therefore, can be considered a reliable calcification proxy.

ii.  Analysis of the seasonal variability of planktic foraminifera calcification revealed important differences between species. *G. bulloides* exhibited peak calcification during winter, *N. incompta* during mid-summer and *G. truncatulinoides* during late summer to autumn.

iii. Interannual analysis suggest that *G. bulloides* and *N. incompta* did not display any significant pattern between 1994 and 2005, on the other hand*, G. truncatulinoides* displays a constant and steady calcification increase over recent years.

iv.  Sediment trap and seabed sediment data comparisons between pre-industrial, post-industrial and recent Holocene assemblages shows that all three species experienced a clear and conspicuous calcification reduction with modern *G. bulloides* populations being 18-24% less calcified and a reduction of 20-27% and 32-40% for *N. incompta* and *G. truncatulinoides*, respectively.

v.   Finally, correlations with environmental parameters and GAM indicate that Optimum Growth Conditions, Sea Surface Temperatures and $CO_3^{2-}$ concentration are the most likely parameters influencing planktic foraminifera calcification in the Northwestern Mediterranean. However, calcification appeared to be species-specific and vary depending on the

time scale studied. This may suggest that other parameters than the ones
studied here may play a role in foraminifera calcification.

As planktic foraminifera represent roughly about 50% of pelagic calcite production
(Schiebel, 2002) in the world's oceans, and therefore, an important component of
the marine carbon cycle, a reduction in the calcification of their shell could induce
important changes in the future carbon cycle with feed-backs on climate. Our results
call for increasing efforts in monitoring planktic foraminifera calcification the
Mediterranean in order to determine if the trends suggested by our data will be
sustained over time.

The Supplement related to this article is available at DOI: 10.17632/4t9x554dwz.1

*Competing interests.* The authors declare that they have no conflict of interest.
*Author contributions.* ASRH, FJS and TMB designed the study. JPT designed Figures 1 and
2 and contributed to data discussion. XDM provided Planier core-top and Lacaze-Duthiers
seabed sediment samples. IC provided the Minorca promontory seabed sediment samples.
NH carried out the $^{14}$C measurements. AH performed the numerical analyses and
contributed to their interpretations. TMB led the sample processing as well as the
microscopy and image analysis, the foraminifera study and wrote the manuscript with
feedback from all authors.
*Acknowledgments.* Authors would like to thank the two anonymous reviewers for their critical
comments that helped improve the manuscript. Authors would also like to thank Blanca
Ausín for her insight on radiocarbon dating and Serge Heussner for the retrieval of the
sediment trap collected within the French national MOOSE program supported by CNRS-
INSU and ALLENVI. This study was funded by the Spanish "Ministerio de Ciencia e
Innovación" through a grant number PRE2019-089091 and through the project RTI2018-
099489-B-100; PID2021-128322NB-I00.

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
