# Peer review of "Calcification response of planktic foraminifera to environmental change in the"

_EGUsphere, 2022_

## Author Response (AR1)

Author's response.

Find here an abstract of the main changes that have been carried out on the manuscript:

- Statistical tests and analysis have been added. First, in order to tests the difference and independence between SBW, area and MBW, a square chi test has been carried out. Also, in order to test the differences between the seasonal/interannual trends and sediment trap and sediment core calcification data, a non-parametric Mann-Whitney variance test has been used. Finally, a General Additive Model has been developed in order to plot seasonal and interannual patterns and for trying to identify which environmental parameters are the most likely to affect calcification on different time scales.
- The $^{14}$C ages of the core-tops and sediment cores has been calibrated to calendar years BP considering the marine reservoir effect and the local reservoir effect. This has been done using the CALIB 8.2 online program. Results showed that 2 of the core-tops previously presented as pre-industrial, are in fact dated from the industrial period. Therefore, discussion around global change and ocean acidification from the pre-industrial time scale has been now adapted in order to integrate these new dates.
- A clearer link has been made with the ecology of the species. For that, a new section has been added in the material and methods focusing on the life cycle of the three species studied here. The seasonal calcification discussion now focuses more on the ecology of the species.
- The pre/post-industrial discussion now also acknowledges other mechanisms than global ocean acidification. Parameters such as temperatures variations, interannual flux-variations and oceanographic changes are now discussed.
- A new section, 5.4., has been added to discuss the variation of the influence of the parameters on different timescales. Here, some parameters non considered in our work are cited as possible mecanisms affecting the impact of environmental parameters on different time-scales.
- The figures have therefore been adapted in order to integrate the changes suggested. Fig.1 now does not show the NWM line. Fig. 2 has now more space and the dot is the decimal separator now. Fig. 3 shows the raw data as suggested. Fig. 4 shows the standard deviation of thew environmental parameters. Fig 5 now shows the interannual raw trend of environmental paramters. Fig. 6 now shows less colors, is easdiert to read and also displays the flux-weighted values for the sediment trap data.
- All data used for this work will be provided according to the Copernicus data policy.
- All the misspelling of the species have been corrected. I.e. *Globigerina bulloides* now reads *G. bulloides*.
- The bibliography has been updated to the Copernicus format.
- DYFAMED site was not well references. Now it ius cited as a program and the correct bibliography has been added.

Now, find the detailed response to every comment from both reviewers.

**RESPONSE TO REVIEWER 1**

First of all, authors would like to thank referee #1 for such a precise and helpful review of our work and its positive feedback on the value of our data. All observations and ideas provided have helped to greatly improved this manuscript. We hope the answers and information provided here would respond to what was demanded.

Here we detail the answers to the questions asked and all the changes and corrections applied to the document in order to integrate the changes suggested by referee #1.
Here, **R1-C** stands for referee #1 **Comment**, and **R1-R** stands for authors **Response**. Referee #1 comments are marked in italic.

As a lot of changes have been proposed, find here a quick abstract of what we think are the main changes that have been suggested:
- Statistical tests and analysis have been added. First, in order to tests the difference and independence between SBW, area and MBW, a square chi test has been carried out. Also, in order to test the differences between the seasonal/interannual trends and sediment trap and sediment core calcification data, a non-parametric Mann-Whitney variance test has been used. Finally, a General Additive Model has been developed in order to plot seasonal and interannual patterns and for trying to identify which environmental parameters are the most likely to affect calcification on different time scales.
- The $^{14}C$ ages of the core-tops and sediment cores has been calibrated to calendar years BP considering the marine reservoir effect and the local reservoir effect. This has been done using the CALIB 8.2 online program. Results showed that 2 of the core-tops previously presented as pre-industrial, are in fact dated from the industrial period. Therefore, discussion around global change and ocean acidification from the pre-industrial time scale has been now adapted in order to integrate these new dates.
- A clearer link has been made with the ecology of the species. For that, a new section has been added in the material and methods focusing on the life cycle of the three species studied here. The seasonal calcification discussion now focuses more on the ecology of the species.
- The pre/post-industrial discussion now also acknowledges other mechanisms than global ocean acidification. Parameters such as temperatures variations, interannual flux-variations and oceanographic changes are now discussed.
- The figures have therefore been adapted in order to integrate the changes suggested.
- All data used for this work will be provided according to the Copernicus data policy.

**Main comments**

*R1-C1: Please show the raw data (individual measurements, not the mean values) and the shell fluxes before showing/analysing the intra and interannual variability. At first I was under the impression that the entire time series was analysed, but I think the authors only analysed selected samples for each species (and not the same samples for each species).*

**R1-R1:** The entire time series has been analyzed, however, not every sample presented an enough amount of planktic foraminifera individuals to carry on the weighing and measurements. We therefore analyzed all sediment trap samples that contained enough *G. bulloides*, *N. incompta* and *G. truncatulinoides*. Sometimes, these 3 species were abundant enough in the same sample, and in some other samples, only one or two of these species were abundant enough. That it's why we could not have and constant calcification pattern for all the three species in the sediment trap, due to the material limitation. The amount of individuals needed for our analysis was described in **section 3.4.** This problem was not encountered in the sediment core material, where individuals from all 3 species were abundant enough and that's why we have the same number of samples analyzed for all 3 species there. The amount of samples for each species analyzed in the sediment trap is displayed in **Table 2**, in the **Total samples** column. On top of that, a table has been added in the supplementary material with all the samples analyzed from the sediment traps from each species, with their corresponding date, SBW, MBW, area and number of individuals.

*R1-C2: To the best of my knowledge Rigual-Hernandez et al. (2012) presented shell fluxes >150 μm. How do the shell fluxes in the specific size fractions analysed here relate to those? In other words, how was the flux-weighting done? Based on size-specific shell flux data, or by assuming that the shell size remained constant and that weights for the calculations of the weighted averages are independent of size?*

**R1-R2:** Yes, Rigual-Hernández et al., (2012) presented shell fluxes from the >150um fraction, and that data is the one used in this study. No new flux data has been generated here and the flux-weighting done for the comparison with the sediment core data has been carried out with Rigual-Hernández et al., (2012) data assuming the uncertainty associated with >150um fraction used. In order to clarify this point, a sentence has been added in section 3.4., it reads: "**The flux data used for these calculations corresponds to the >150 um for each species after Rigual-Hernández et al., (2012).**"

*R1-C3: Please provide the depths of the analysed sediment samples. I assume that the dated sediments have also been analysed, but it's not obvious from the text. Why were the only replicate analyses done on the core top (PLA CT)? What was the spread?*

**R1-R3:** The depth of the sediment samples analyzed as well as the sediment samples dated are now inserted in **Table 1**. A sentence has been added at the end of section 3.6.,

it reads: **"Planktic foraminifera present in the dated samples that were not selected for radiocarbon dating were also analyzed following the methodology described previously"**.

*R1-C4: Details on the statistics are scarce. What tests were used for what reason? Are they applicable (the data appear log-normally distributed) and are predictor and response variables really independent (Fig. 2, shows area against area normalised weight).*

**R1-R4:** Corrected according to suggestion. **Section 3.6**. now describes only the statistical analysis. Concerning the difference between SBW and MBW, Beer et al., (2010) methodology was followed. The main idea was just to prove that our MBW values were not correlated with the morphometric parametric they aim to be corrected for. To further support this observation, we performed a chi2 independence test and the results are discussed in **section 4.1.** However, and as asked in the following comments, statistical tests have been added to the study. In particular, to test the difference between MBWs from the sediment trap and the sediment cores, a non-parametric Mann-Whitney test has been used. This non-parametric test assesses whether two sampled groups are likely to derive from the same population by comparing the medians of the different datasets. A p-value of <0.05 has been used in order to consider the median of two datasets significantly different. This test has also been used for the seasonal and interannual trends.

Additionally, a description of the the GAM model requested by reviewer #1 in the following comments has also been included in **section 3.6.**

*R1-C5: Along the same lines, what is the meaning of the error bars in the various figures? In Fig. 3 it seems that they reflect the uncertainty (standard deviation?) of the sample averages for each month. My most important point is that this hides a large part of the variability (see Table 2) and I recommend to show the individual values (i.e. for each measured shell, normalised to average weight). In addition, how were samples assigned to a month, by start date, mid date, or by taking into account the collection interval? And are the mean and the standard deviation appropriate summary statistics for these data that are far from normally distributed? The same applies to Fig. 4? And in the latter figure, are the mean values flux-weighted?*

**R1-R5:** Corrected according to suggestion. The error bars, which reflected the standard deviation, from figure 3 and figure 4 have been removed. Also, now the individual values available from each month and each year are now plotted, along with the already plotted mean values. The samples were assigned into their corresponding month and year taking into account the exact collection date. This is now specified in **section 4.2.** before **Figure 3** and in the caption of **Figure 4**.

**R1-C6**: *Why do none of the environmental variables have error bars? This is especially relevant for the analysis of the seasonal patterns.*

**R1-R6**: Corrected according to suggestion. Error bars have been added. As specified below and in the GAM description, only the carbonate system parameters have been left in the seasonal and interannual figures as they could not have been included in the GAM.

**R1-C7**: *What is the rationale to infer linear trends in Fig. 4? And are the calcification intensity values flux-weighted? And how much of the variability is due to changes in the shell flux?*

**R1-R7**: Linear trends were initially included in Figure 4 to illustrate the evolution of the calcification of the three species analyzed during the 12-year record. In the case of *G. bulloides* and *N. incompta* the trend was flat, excepting the last 2-3 years. In the case of *G. truncatulinoides* our data suggests a clear increasing pattern in shell calcification that was clearly illustrated with the linear trend.

However, we agree with reviewer #1 in that the calcification response does not necessarily has to be linear and consequently the linear trends were removed in the new version of the manuscript. In order to improve the interpretation of the interannual variability, GAM has been employed. As described below (see R1-R9 next section), the GAM could not include the carbonate system parameters, therefore these parameters have been maintained in this figure. These values are not flux-weighted, as they just aim to show a measurement of the degree of calcification during the high productivity period of each year.

**R1-C8**: Please provide the data in accordance with the Copernicus data policy: https://www.biogeosciences.net/policies/data_policy.html (preferrably also for review)

**R1-R8**: Corrected according to suggestion. All supplementary data and figures used in the discussion will be provided according to Copernicus data policy. Also, the environmental data, raw calculations of SBW, morphometric data, MBW calculations and sediment core weighing will also be provided.

**Additional points:**

**R1-C9**: *Separate analysis of intra and inter-annual variability. The authors first analyse the intra- annual variability in calcification intensity and then proceed with the inter-annual. What this means is that some of the interannual variability is incorporated in the composite seasonal cycle, perhaps obscuring or suggesting trends that are not there. Why not model the calcification intensity as a function of month/day of the year and time? This will allow obtaining more meaningful estimates of the seasonal cycle and the long-term trend and can for instance be done using GAMs.*

**R1-R9:** Authors agree with this observation. In order to model both calcification patterns, a GAM has been designed as suggested. This GAM has modeled the variations on a monthly basis and has extracted the seasonal and interannual cycle for both the calcification and the environmental parameters selected. Details about the GAM development can now be found in **section 3.6**. Previous correlation data are still presented in a reduced format (see comment and response R1-R35 in the next section). This correlation data have been used in the GAM in order to constrain the relation between the parameters and refine the analysis. This GAM has been used for both the seasonal/interannual patterns and the correlations with the environmental parameters. Due to the limited availability of carbonate parameters data for our study period, the GAM could only be performed with the Planier data (MBWs, fluxes, chlorophyll-a and SST), and the environmental data from DYFAMED site (Salinity and nutrient concentration). The carbonate system parameters, as excluded from the GAM, are therefore presented as a combination of correlations and linear models. The results and figures generated by the GAM are now included in the new version of the manuscript while details of the calculations can be found in the Supplement.

*R1-C10: Significance of the trends/patterns. Irrespective of the approach the authors take (although I recommend modelling the seasonal and interannual data together) the authors need to establish whether or not the trends/differences are significant. I am not convinced this is the case for any of the time scales considered (seasonal, interannual and decadal/centennial; with the possible exception of G. truncatulinoides on intra-annual and centennial time scales) and the authors run the risk of trying to explain noise (and derive meaningless conclusions) in the sections where they attempt to explain what drives variability in calcification intensity.*

**R1-R10**: A Mann-Whitney test has been carried out to test the differences in the medians for seasonal/interannual trends but also for the sediment trap and core-top datasets.

*R1-C11: Drivers of variation in calcification intensity. I agree with the authors that identifying the exact drivers of calcification intensity is not trivial because of collinearity in the environmental variables. However, there are ways to better handle this problem than assessing the correlation of each variable individually. I can think of multiple linear regression or more flexible GAMs.*

**R1-R11:** See R1-R9 in this same section of comments.

*R1-C12: Differences among species. One of the most interesting observations is that different species show different patterns in the calcification intensity (the latter still needs to be demonstrated and I suspect that only G. truncatulinoides shows a significant pattern on seasonal time scales). Right now it is hidden in the text, but it would be interesting to discuss in greater depth and with more focus why this might be the case and make a clearer link to the ecology of the species. The relationship between calcification intensity and shell flux is particularly intriguing in this respect and the authors go some way to discussing this for*

*truncatulinoides (allocation of energy), but not for the other species. I recommend that they do.*

**R1-R12:** Authors appreciate the point made by the reviewer and agree with this observation. Consequently, a new section has been added to the material and methods, **section 3.3**. entitled: "**Ecology and life cycle of *G. bulloides*, *N. incompta* and *G. truncatulinoides*"**. On top of that, a clearer link has been established with the ecology of the species. In the case of *G. bulloides* and *G. truncatulinoides*, the life cycles and the calcification processes are more or less well defined, however, this is not the case for N. incompta, therefore the discussion surrounding this species is more speculative. This aspects are now discussed in the OGC impact on calcification in **section 5.1.**

*R1-C13: Related to this is the concept of optimum growth conditions and whether shell flux (mortality in fact for sediment traps) is a good measure (perhaps not), which should be discussed in greater depth. Perhaps the relative abundance of a species is a better indication of optimum growth conditions? Or the first derivative of the shell flux/abundance as the fastest growth occurs when the population size increases most.*

**R1-R13:** Authors agree that the OGC (Optimum Growth Conditions) are a complex index that should be interpreted with caution. In our study we first considered the relative abundances, however, the problem with them is that they are dependent on each other, implying that the increase in one species relative contribution means that other species abundance decreases. Therefore, the use of relative abundance could could lead to missinterpretations of the results. In turn, foraminifera fluxes are independent of each other. That's the main reason we used the >150um fluxes from Rigual-Hernández et al., (2012). In our paper, we first consider shell fluxes as OGC, then the chlorophyll-*a* as OGC (de Villiers et al., 2004) and finally compare this approach with the one that considers the OGC as nutrients (Schiebel et al., 2001). Therefore, the aim of our work is not about which environmental factors determine OGC, but using the main factors reporting in the literature to influence OGC to assess their possible role in calcification. However, we acknowledge that our approach was not clear enough in the first version of the manuscript and, therefore, our strategy to assess the impact of OGC on foraminifera calcification has been clarified in the new version of the discussion. In particular in **section 5.1.** It reads: "**Here, we first approach seasonal shell calcification by considering the Optimum Growth Conditions (OGC). Previous studied have defined these conditions on a wide variety of ways: abundance of foraminifera, the chlorophyll-a concentration and even nutrients concentration (de Villiers et al., 2004; Schiebel, 2001; Schiebel and Hemleben, 2017). Therefore, we aim to explore the impact of these parameters as OGC on the shell calcification."**

*R1-C14: Differences among time scales. Another intriguing finding is that the environmental drivers of calcification intensity seem to vary with time scale. I am not sure if this is a robust finding (see above), but at face value the calcification intensity of truncatulinoides appears*

*positively correlated with temperature and carbonate ion concentration on seasonal time scales, negatively correlated on interannual time scales and on longer time scales positively with temperature and negatively with CO32-. If real, why could this be the case? The authors hardly touch one this, but it is interesting as it suggests that different mechanisms than considered might be responsible.*

**R1-R14:** Authors agree with this observation and the fact that other mechanisms could be involved. Therefore a small discussion for each species has been added around the variability of the impact of environmental parameters on different time scales. (). This discussion has been added as a new section, **section 5.4**., it reads: "**environmental parameters influence across different time scales".** Other parameters such as the ecology of the species on different time scales and oceanographic changes are discussed.

***R1-C15****: Ages of the core tops. There are several issues with paragraph 4.4 where the calcification intensity from the sediment trap time series is compared to the core top data. First of all, it is unclear what is actually compared. Fig. 6, which is referred to in this section, shows the mean (I presume) values for each sample (cup or depth), not the flux-weighted calcification intensity from the sediment trap time series and the core top data as is suggested in the text (which would mean 1 data point from the trap and one for each core top). This needs to be clarified. It makes of course most sense to compare what the authors write (but they need to explain how exactly they derive the flux-weighted average, right now it is unclear if this is based on the monthly averages or on the entire time series), but the figure shows something else. Notwithstanding the calculation, it is interesting that the differences are so consistent, I think the authors should emphasise that more. Second, the authors have not calibrated the 14C ages to calendar years. This means that the age/durations mentioned in the text are simply wrong. The authors need to correct for the local reservoir effect and calibrate the ages before they can make any statements about ages or rates. I realise that this does not affect the difference in calcification, but it is nevertheless important for the interpretation (as the core tops might on average not be post-industrial, but see below). Thirdly, the 14C ages are average values only. Due to time integration within each sample and because of bioturbation the real ages of the foraminifera vary (Dolman et al., 2021; Lougheed and Metcalfe, 2021). The authors need to acknowledge this in their interpretation.*

**R1-R15:** We appreciate all the important insights made by the reviewer. Figure 6 shows all the mean values for the sediment trap, however, as written in the text, what is compared in the main text are the flux-weighted values of the 12-year long sediment trap time series with the data from the sediment cores. Following reviewer's suggestions Figure 6 has been modified in order to show the sediment trap flux-weighted values. Secondly, authors have revised this section of the work and calibrated the 14C ages to calendar years. The details of the calibration displayed here can now also be found in **section 3.7**. Radiocarbon ages from the 3 sediment cores were calibrated using the CALIB program (Stuiver and Reimer, 1993) and using the Marine20 curve, which applies a marine reservoir correction of 550 14C

years. Additionally, a -165 ± 93 years local reservoir was also considered (ΔR) (Stuiver and Braziunas, 1993). This ΔR was calculated as the average of the nearest 8 points to the sample location from the Marine Reservoir Correction database, whose values have already been corrected for the Marine20 curve. After these corrections, the sample from Menorca (MR 3.1.A 14-14.5 cm) displayed a date of 1557 (rounded to 1560) calendar years BP. So, samples from this setting can be considered pre-industrial. In regard to the calibration on the core-top samples from Planier and Lacaze-Duthiers sites (0.5-1cm), they were also corrected considering the local reservoir correction : 490(± 60) and 460 (±60), respectively. Based on these results the margin of the datations do not allow us to determine with certainty if these samples are pre-industrial. In order to give a rough idea of the time span they could cover, the most recent age accepted for calibration by CALIB 8.2 (i.e. 603 14C years BP) has been used as (and the results are shown in the Supplement) has been used as a reference value (227 cal years BP with a very low possibility of being posterior to 1950 AD) with the same marine and local reservoirs previously considered. Therefore, we can consider that out samples could be dated anywhere between 227 cal years BP and 1950 AD as some bomb $^{14}$C has been found. All of this datations are now available in **Table 1** and in **section 3.7**. As stated before, this does not affect calcification and the trends are the same: a reduction of calcification for the 3 species in the sediment cores compared to the sediment traps. Therefore, we have rewritten the paragraph dedicated to the dating of Planier and Lacaze-Duthiers samples. As a result of all these changes, the new version of the discussion is orientated around global climate change and ocean acidification but it also considers other mechanisms that could be responsible for this calcification decrease. It now also includes the samples that cannot be considered pre-industrial but industrial times samples as a discussion of the changes in calcification during the late industrial to recent Holocene.

*R1-C16: Attribution of difference with sedimentary shells. With regards to the line of argumentation to exclude the effect of dissolution I disagree with the authors, particularly on the third point ("if partial dissolution was to take place here, MBWs from the seabed sediments and core tops would be lighter than the ones from the sediment traps" L762-764). (Seafloor) Dissolution tends to affect the thinnest shells first, leaving the sediment enriched in thick/more heavily calcified shells (Berger, 1970 and many more studies since), thus exactly what the authors observe. The difference in calcification intensity thus seems to be in agreement with dissolution affecting the sedimentary shells instead of providing evidence against it.*

**R1-R16:** Authors understand the point made here and agree with the bibliography provided. Therefore, the hypothesis regarding the possible influence of selective dissolution on the sea floor sediments is not discarded in the new version of the manuscript. This hypothesis is now presented as a possible explanation for the differences between sediment traps and sediment assemblages. However, while the possibility of selective dissolution is not ruled out, this idea is presented as less probable than the other hypothesis because there are several lines of evidence that still suggest that carbonate preservation is optimum in the

sediment samples, including:the carbonate supersaturation in the Mediterranean sea (Álvarez et al., 2014) and depth of the sediment cores studied (990 to 2117m) as well as the negligible dissolution of calcareous nannoplankton in the transit from the trap to the floor (Beaufort et al., 2007; Moy et al., 2009), make the authors think that dissolution, although not completely ruled out, seems unlikely here. A paragraph has been added at the of third paragraph of **section 5.3.,** it reads: **"However, it has been described that when dissolution takes place, it affects the thinnest shells first (Berger, 1970), and only the heaviest and more calcified specimens remain. In our study, the specimens from the sediment trap were lighter that the ones from the sediment cores, therefore, this is important to acknowledge as, although unlikely, dissolution cannot be completely ruled out."**

*R1-C17: Similarly, the fourth argument ("SEM observations of all 3 species in samples from the sediment traps showed no sign of dissolution and foraminifera were well preserved" L765) is also unconving. Shells spend orders of magnitude more time on the sediment-water interface than in the water column and dissolution in the water column is therefore less of a problem. The authors therefore need SEM images of the sedimentary shells to prove that they were not affected by dissolution. Ideally, they also include SEM pictures of cracked sedimentary shells to demonstrate that the cleaning was sufficient to remove any sediment from within the shells.*

**R1-R17:** Authors agree with the point made by the reviewer and will provide additional SEM images of sedimentary shells and broken shells in order to demonstrate the effectiveness of the cleaning technique.

*R1-C18: In addition, can the authors rule out that the difference in calcification intensity is not due to changes in the seasonality of the flux (because of the seasonal variability in calcification intensity, it is after all possible to change the flux-weighted mean calcification intensity by just changing the flux; cf de Moel et al., 2009) or due to addition of calcite below the depth of the trap (G. truncatulinoides)? So, even if the difference in calcification intensity between the core top and the sediment trap shell is significant (which needs to be demonstrated first), there are still other options than ocean acidification (based on the seasonal pattern, temperature seems an obvious candidate) that could at least in theory explain the difference. The authors therefore need to provide more convincing arguments to support their conclusions.*

**R1-R18:** Authors appreciate the point made by the reviewer. in order to prove the difference between the different datasets (sediment trap vs sediment cores  a non-parametric Mann-Whitney has been included in **section 3.6.** This test assesses the difference between the medians of the different datasets (see R1-R4 in the previous section for more details). Authors agree that changes in other environmental parameters aside from ocean acidification could also account for the observed differences between datasets. Therefore, the end of **section 5.3.** now includes a paragraph that discusses the possible role of

changes in other environmental factors (such as SSTs, changes in fluxes and on the oceanographic setting), that could also account for the observed reduction in shell calcification in the three species.

*R1-C19: The authors may want to include the paper by Weinkauf et al (Weinkauf et al., 2016) in their discussion.*

**R1-R19:** This manuscript is now mentioned and discussed in the manuscript as suggested by the reviewer.

**Specific comments**

*R1-C20: The title does not seem consistent with the objective as specified in the first sentence of the abstract.*

**R1-R20:** Corrected according to suggestion. Now the abstract reads: "**The aim of this work is to investigate the response of planktic foraminifera calcification in the northwestern Mediterranean Sea on different time scales across the industrial era**."

*R1-C21: L59: the industrial period started about 170 years ago (although somewhere else in the text the authors suggest it started around 1800 CE).*

**R1-R21:** Corrected according to suggestion. Now it reads "**defined according to Sabine et al., (2004) from 1800 and therein**".

*R1-C22: L60: unprecedented for what time frame?*

**R1-R22**: Corrected according to suggestion. Now it reads "**has caused an increase in carbon dioxide"**

*R1-C23: L77: Jonkers et al did not discuss ocean acidification.*

**R1-R23**: Corrected according to suggestion. Reference deleted.

*R1-C24: L91: "bearing" not "wearing". I think the word depending here is also ambiguous, it may suggest that ecology and feeding strategy evolved first and that the species added symbionts or spines. If anything, I think it was the other way around. Better phrase this neutrally.*

**R1-R24**: Authors agree with this observation. Corrected according to suggestion.

*R1-C25: L113-124: Is this section needed here? A lot of the information is also presented in the section "study area"*

**R1-R25**: As this information was not exactly present in the "study area" section, it has been moved there. This part in the "Introduction" section has been shorted and now it reads "**The Mediterranean Sea is a semi-enclosed sea with a high saturation state for calcite (Álvarez et al., 2014). It is often considered as a "miniature ocean" and a "laboratory basin" (Malanotte-Rizoli, 2010), which makes it a valuable zone to study marine calcifiers shell calcification processes.**"

*R1-C26: L124: "MedECC"?*

**R1-R26**: MedECC: Mediterranean Experts on Climate Change. Corrected according to suggestion, now the acronym is described in full text. Reference and bibliography updated.

*R1-C27: L129: insert "can" before "provide", short (< 1 year) deployments cannot provide annually integrated fluxes.*

**R1-R27**: Corrected according to suggestion.

*R1-C28: L146-193: can this not be condensed to the information that is relevant to the study?*

**R1-R928**: This section has been condensed but authors felt it was necessary to keep a few sentences about the Northern Current formation. Now it reads :" **The Mediterranean is a semi-enclosed sea and it is considered a concentration basin… The NC largely controls the circulation all over the western and northwestern part of the Mediterranean Sea, including the Gulf of Lions (Millot, 1991) and the Balearic Sea (Figure 1a)**". See comment R1-C6 and R1-R6 for the section that has been moved here.

*R1-C29: Fig. 1: the line representing NWM lacks an arrow to indicate the direction of the flow. It also seems oddly placed as if it connects Minorca with Sicily.*

**R1-R29**: The NWM line was meant to show the limits of what is considered the NW Mediterranean. It did not represent a flow or current. In order to avoid misunderstandings, this line has been removed in the new version of the manuscript.

*R1-C30: L206: "the characterisation and quantification" ... "were analysed" reword*

**R1-R30**: Corrected according to suggestion. Now it reads: "**Planktic foraminifera fluxes for the 1993 to 2006 period were documented by Rigual-Hernández et al., (2012).**"

*R1-C31: Table 1: provide reservoir age (R or delta R) and calibrated age (include details on calibration in the methods)*

**R1-R31:** Corrected according to suggestion. Details on the calibration can now be found in **section 3.7.**

*R1-C32*: *L237-241: please be clearer that the number of unique samples processed (Table 2) is lower. Also explain why species were analysed in size fractions, it is not immediately obvious since the size is also measured, and that this may lead to an underestimation of the variability in calcification intensity in the entire population of the planktonic foraminifera species.*

**R1-R32:** Corrected according to suggestion. Now it reads:" **However, these numbers represent the total of samples analyzed but unique samples number is lower, as not all the sediment trap samples presented the three species in high enough numbers to perform the picking. The species were analyzed in size fractions in order to estimate the efficiency of sieve fractions and the impact of size and morphometric parameters on the foraminifera weight and calcification,**".

*R1- C33:* *L242-247: please explain why this number of shells is sufficient to characterise the variability/mean within a sample.*

**R1-R33:** Corrected according to suggestion. Now it reads*:* **The lowest number of individuals selected per sample was 5 in order to maximize the number of samples available for our study. According to Beer et al., (2010), the higher the number of individuals, the more reliable SBWs are. Here we aim to compare SBW results with a measured weight technique. Measured techniques are acknowledged to be reliable with a lower number of individuals, therefore a minimum of 5 individuals were selected in order to compare the two techniques.** *".*

*R1-C34*: *L267-270: were the morphometric analyses done on the same shells that were weighted (i.e. the 10-30 shells in specific size fractions)?*

**R1-R34:** Yes, they were. Added a sentence according to suggestion. It reads **:" These measurements were carried out on the same shells that were weighted."**

*R1-C35*: *L275: I understand the goal of this normalisation, but please describe better what the difference is between "mean parameter size fraction" and "mean parameter sample". And what is the advantage of this method compared to the area density (they seem highly correlated).*

**R1-R35:** Corrected according to suggestion. Now it reads: "**Size fraction" accounts for the mean of the parameter (area or diameter) measured in all the sites studied, while "sample" accounts for the mean of the parameter in the sample being measured. The advantage of these measurements is that the resulting MBW is being given with a weight unity (µg), which makes comparable to other studies (Beer et al., 2010).**."

***R1-C36****: L288: SeaWiFS started in 1997, please mention this and explain why you analyse composites instead of analysing each sample using the corresponding environmental variable.*

**R1-R36:** Corrected according to suggestion. Now it reads: "**SeaWiFS measurements started in 1997 and were used due to the lack of direct chlorophyll measurements in the Planier sediment trap deployment site.**"

***R1-C37****: Table 2: is standard deviation meaningful for data that are distributed like this?*

**R1-R37**: Authors beleive this parameter is meaningful and appropriate here because it allows a comparison of the variations of the morphometric parameters measured between the different sites. It is also a supplementary argument to show that the use of a narrow size-fraction for analyzing the calcification of planktic foraminifera.

***R1-C38****: L376: it would be good to explain the rationale for testing this better and earlier. Size and weight are obviously correlated and it is not a priori clear why that is an issue.*

*The names MBWarea and MBWdiameter are a bit confusing, especially since the unit is microgram. It is interesting to see that there is still a size weight relationship within a narrow size fraction, but not entirely surprising. What are the implications?*

**R1-R38:** Corrected according to suggestion. A paragraph has been added at the end of **section 3.4.**, it reads: "**Correlations between SBW and MBW$_{area}$ against area are displayed in Figure 2. The reason of this comparison is to show the relation between size and weight. In order to avoid the impact of having the bigger specimens displaying the heaviest weight and impacting the mean weight (therefore calcification indicator) of the sample.**". Concerning the implications of having some influence of the area on the MBWarea is not surprising (as stated in the text). However, as the $r^2$ showed are very low, this influence can be considered minimal compared to the influence of area on the SBW. Therefore the authors consider that it is a better calcification indicator.

***R1-C39:*** *Fig. 2: please try to make this figure clearer. Allow more space between the subplots and show the axes next to the points. Make sure that the decimal separator is a point and not a comma. Add space between genus abbreviation and species name.*

**R1-R39:** Corrected according to suggestion.

***R1-C40:*** *L409-411: and variability within the size range.*

**R1-R40:** Added to text according to suggestion.

***R1-C41***: *L417-418: an R2 of 1seems an artefact. It would only occur if all shells have the same shape, which I doubt.*

**R1-R41:** It is exponential correlation, therefore area and diameter are expected to be perfectly correlated. It has been done as a quality control for the morphometric measurements. In order to avoid confusion, this sentence has been removed.

***R1-C42***: *L431: "Mean annual MBWarea and roA values were calculated ... to illustrate the seasonal variability" does not make sense. It seems that monthly values were calculated. And why the average? It does not seem a robust indicator.*

**R1-R42:** Corrected according to suggestion. The average value has been taken due to the number of values available.

***R1-C43***: *Fig. 3: (in addition to the comments above): use point instead of comma, add space in species name, add number of observations per month. Why are there no error bars for roA? (and what is the advantage of showing both metrics?)*

**R1-R43:** Corrected according to suggestion. The number of observations per month as well as the total observations across the time span have been added. The roA plots were deleted because they are not discussed in the rest in the study.

***R1-C44***: *L445-447: These two sentences mean the same (and I do not agree that all species show clear patterns at all).*

**R1-R44:** Corrected according to suggestion. Now it reads: "**The seasonal variations in shell calcification differ according to the species**".

***R1-C45***: *L469: are these values flux-weighted?*

**R1-R45:** No, these values are not flux-weighted. Here we aim to just present the data from the different years in order to see if the calcification varied on an inter-annual scale. This is know acknowledged in the manuscript, at the beginning of section 4.3., it reads: "**As our aim is to present the raw interannual calcification trends, these values have not been flux-weighted**".

***R1-C46:*** *Fig. 4: comma and space. It is unclear to me why the carbonate system parameters have not been averaged. The linear trends do not make sense, especially for the nutrients and salinity.*

**R1-R46:** Corrected according to suggestion. This figure has been modified and simplified, keeping only the interannual trends. As suggested in R1-R1 in previous section, a GAM has been carried out and, in order not to be repetitive with the plots, we took all the environmental

parameters out of this plot. As donne in the previous figure, the number of observations per year has been added.

*R1-C47: L481: knit-picking, but there can only be one minimum value.*

**R1-R47:** Corrected according to suggestion. Now it reads: "**Lower values".**

*R1-C48: L485: the increase in calcification intensity is not constant.*

**R1-R48:** Corrected according to suggestion. Now it reads: "**with an overall steep calcification increase throughout the record**".

*R1-C49: L491-492: I am sure that the other environmental variables also showed interannual variability. Why treat the carbonate system differently?*

**R1-R49:** Corrected according to suggestion. At first they were treated different due to the fact that they were only available for 2 periods. The paragraph has been reworked in order to treat all parameters the same. It reads: **"All environmental parameters showed variations across the years. Sea Surface Temperatures… Between the 2 periods for which direct on-situ carbonate system parameters measurements were available".**

*R1-C50: L495-496: This sentence is irrelevant here in the results section. Move to discussion if relevant at all.*

**R1-R50:** Authors agree with the fact that the sentence is irrelevant and therefore, it has been deleted.

*R1-C51: L500-501: this sentence seems to come too early, you have not yet established that there is a reduction in calcification intensity. In addition, the figure does not allow to infer much about any Holocene trends since there is no time axis.*

**R1-R51:** Corrected according to suggestion. Now it reads: **"Foraminifera weights analyzed in core tops and sediment cores from the NW part of the Mediterranean (Figure 6) and radiocarbon dating allowed a further insight on foraminifera calcification during the Holocene."**

*R1-C52: L502: how was the flux-weighting done? Using the monthly averages, or the observed data? If the latter, is it not biassed to the times (fluxes) when the observations were made?*

**R1-R52:** The flux-weighting is described in section 3.4. and is cited in the text. It reads: "**MBWs were flux-weighted. Mean monthly MBWs values from each species were**

**multiplied by the corresponding mean monthly flux and then divided by the total annual flux of the corresponding species**".

*R1-C53: L506: "in the last 489 years" is not correct, see comments above.*

**R1-R53:** Corrected according to suggestion. The sentence has been deleted.

*R1-C54: Table 3: what does seasonal mean in this case? Monthly, three-monthly (which months?)? How was significance determined? And is that appropriate? Why provide the entire matrix, it contains a lot of redundant information. And what is the purpose of correlating everything with everything (how relevant is it to show that phosphate and carbonate ion concentration (use correct notation) are correlated)? Simply show what is relevant and discussed.*

**R1-R54:** Seasonal means monthly in this case, and it has been added in the caption of **Table 3**. Significance was determined using a p-value of <0.05 as stated **Table 3**. Caption, it reads: **"Significant correlations (p<0.05) are set in bold." Table 3** has now been reduced to show only the relevant and discussed values.

*R1-C55: L558-561: this is interesting. Perhaps add some information about the life cycle of the different species here to discuss why truncatulinoides shows a unique pattern.*

**R1-R55:** Corrected according to suggestion. A paragraph has been added, it reads: "**As described previously (see section 3.3), this species life cycle is complex and it migrates through different depths of the water column. It is thought to reproduce once a year in winter in subtropical waters and it has been speculated that nutrient availability and the lack of predation could explain this strategy. Therefore, our data, that displayed a heavier calcification in autumn/winter for this species, could show adults that have spent time in deeper waters developing a thick calcite crust come to shallower waters in late autumn to winter and reproduce. This coincides with the period in which the other species are less present and therefore, could allow *G. truncatulinoides* to reproduce due to the lack of competition and predation**". This paragraph has also been extended in order to discuss the ecology of the remaining species (see comment R1-R12). On top, a section has been added about the ecology of all three species: **Section 3.3.**

*R1-C56: Fig. 5: commas. Not all environmental variables are from the DYFAMED site (or the caption of table 2 is incorrect).*

**R1-R56:** Corrected according to suggestion.

*R1-C57: L591-594: the use of the term OGC seems confused. Obviously, the niche of the species is multi-dimensional, i.e. the species are likely to have food and other preferences*

*(temperature for instance), but it is not likely that OGC are linearly related to any environmental variable (e.g. it may be too hot for a species). OGC occur within a range of environmental variables, and there is hence no "proxy" for it, they can only be described (e.g, OGC are between x and y). Without establishing what OGC actually are, it is therefore difficult to use them as a predictor of growth/calcification intensity (L665).*

**R1-R57:** In this study, OGC have been first described as species fluxes and then as chlorophyll- a concentration and we aimed to see the differences between these 2 approaches. In this particular case, OGC are defined as chlorophyll-a concentration, but they are discussed as covariation with nutrients could lead to think that the latter are a better OGC indicator. However, a sentence clarifying that the OGC proxy has to be taken with care has been added in order to avoid confusion. It reads: **"Although here we have first described the OGC as species fluxes and then as the chlorophyll-a concentration, it is important to remember that the niche and favorable conditions meant to be described by the OGC for each species are multi-dimensional"**.

*R1-C58: L617-618: if this is true, what would the sensitivity be (how much less calcification with how much more phosphate) and how does this compare with the studies cited?*

**R1-R58**: Corrected according to suggestion.

*R1-C59: L666: what aspects of seasonal changes?*

**R1-R59:** Calcification seasonal changes. Corrected according to suggestion.

*R1-C60: L685: presumable Fig 4. Are the trends significant?*

**R1-R60:** Yes. Corrected according to suggestion.

*R1-C61: L716: figure 4 instead?*

**R1-R61:** Yes. Corrected according to suggestion.

*R1-C62: L718-720: "the recent SST decrease". Can this not be tested explicitly?*

**R1-R62:** As this is just a theory around an oceanographic process, it has not been tested explicitly as is not the main focus of the paper.

*R1-C63: Fig 6: commas. What is compared (see above). The different colours for the species are redundant here as they are in separate graphs. Consider changing them and give more distinct colours to the sites to improve the clarity.*

**R1-R63:** Corrected according to suggestion. The colours have been changed. See coments above for the data that has been changed.

*R1-C64: L797: see Bird (Bird et al., 2017) for bulloides.*

**R1-R64:** Ressource included in the discussion.

*R1-C65: L859: what is in the supplement and where can it be found?*

**R1-R65:** The supplement will be furnished in the next round of discussion.

REFERENCES:

Álvarez, M., Sanleón-Bartolomé, H., Tanhua, T., Mintrop, L., Luchetta, A., Cantoni, C., Schroeder, K., and Civitarese, G.: The $CO_2$; system in the Mediterranean Sea: a basin wide perspective, Ocean Sci., 10, 69–92, https://doi.org/10.5194/os-10-69-2014, 2014.

Beaufort, L., Probert, I., and Buchet, N.: Effects of acidification and primary production on coccolith weight: Implications for carbonate transfer from the surface to the deep ocean: OCEANIC CARBONATE TRANSFER, Geochem. Geophys. Geosyst., 8, n/a-n/a, https://doi.org/10.1029/2006GC001493, 2007.

Beer, C. J., Schiebel, R., and Wilson, P. A.: Technical Note: On methodologies for determining the size-normalised weight of planktic foraminifera, Biogeosciences, 7, 2193–2198, https://doi.org/10.5194/bg-7-2193-2010, 2010.

Bergamasco, A. and Malanotte-Rizzoli, P.: The circulation of the Mediterranean Sea: a historical review of experimental investigations, Advances in Oceanography and Limnology, 1, 11–28, https://doi.org/10.1080/19475721.2010.491656, 2010.

Berger, W. H.: Planktonic Foraminifera: Selective solution and the lysocline, Mar. Geol., 8, 111–138, 1970.

Heaton, T. J., Köhler, P., Butzin, M., Bard, E., Reimer, R. W., Austin, W. E. N., Bronk Ramsey, C., Grootes, P. M., Hughen, K. A., Kromer, B., Reimer, P. J., Adkins, J., Burke, A., Cook, M. S., Olsen, J., and Skinner, L. C.: Marine20—The Marine Radiocarbon Age Calibration Curve (0–55,000 cal BP), Radiocarbon, 62, 779–820, https://doi.org/10.1017/RDC.2020.68, 2020.

Millot, C.: Mesoscale and seasonal variabilities of the circulation in the western Mediterranean, Dynamics of Atmospheres and Oceans, 15, 179–214, https://doi.org/10.1016/0377-0265(91)90020-G, 1991.

Reimer, P. J. and Reimer, R. W.: A Marine Reservoir Correction Database and On-Line Interface, Radiocarbon, 43, 461–463, https://doi.org/10.1017/S0033822200038339, 2001.

Rigual-Hernández, A. S., Sierro, F. J., Bárcena, M. A., Flores, J. A., and Heussner, S.: Seasonal and interannual changes of planktic foraminiferal fluxes in the Gulf of Lions (NW Mediterranean) and their implications for paleoceanographic studies: Two 12-year sediment trap records, Deep Sea Research Part I: Oceanographic Research Papers, 66, 26–40, https://doi.org/10.1016/j.dsr.2012.03.011, 2012.

Sabine, C. L., Feely, R. A., Gruber, N., Key, R. M., Lee, K., Bullister, J. L., Wanninkhof, R., Wong, C. S., Wallace, D. W. R., Tilbrook, B., Millero, F. J., Peng, T.-H., Kozyr, A., Ono, T., and Rios, A. F.: The Oceanic Sink for Anthropogenic CO$_2$, Science, 305, 367–371, https://doi.org/10.1126/science.1097403, 2004.

Schiebel, R., Waniek, J., Bork, M., and Hemleben, C.: Planktic foraminiferal production stimulated by chlorophyll redistribution and entrainment of nutrients, Deep Sea Research Part I: Oceanographic Research Papers, 48, 721–740, (Schiebel et al., 2001), 2001.

Schiebel, R. and Hemleben, C.: Planktic Foraminifers in the Modern Ocean, Springer Berlin Heidelberg, Berlin, Heidelberg, https://doi.org/10.1007/978-3-662-50297-6, 2017.

Stuiver, M. and Braziunas, T. F.: Modeling Atmospheric $^{14}$C Influences and $^{14}$C Ages of Marine Samples to 10,000 BC, Radiocarbon, 35, 137–189, https://doi.org/10.1017/S0033822200013874, 1993.

Stuiver, M. and Reimer, P. J.: Extended $^{14}$C Data Base and Revised CALIB 3.0 $^{14}$C Age Calibration Program, Radiocarbon, 35, 215–230, https://doi.org/10.1017/S0033822200013904, 1993.

de Villiers, S.: Optimum growth conditions as opposed to calcite saturation as a control on the calcification rate and shell-weight of marine foraminifera, Marine Biology, 144, 45–49, https://doi.org/10.1007/s00227-003-1183-8, 2004.

**RESPONSE TO REVIEWER 2:**

The authors would like to thank referee #2 for taking the time to review the manuscript and for its positive feedback. We acknowledge that the inputs and suggestions provided have really helped to improve the quality of this work.

Here we detail the corrections that have been applied and changes that have been done in order to integrate referee #2 suggestions. Here, **R1-C** stands for referee #1 **Comment**, and **R1-R** stands for authors **Response**. Referee #1 comments are marked in italic.

*R2-C1: Abstract:*

*I found it to be a bit too methodological, too much information to move into the materials and methods (i.e. As the traditionally used sieve fractions method is considered unreliable because of the effect of morphometric parameters on the foraminifera weight, we measured area and diameter to constrain the effect of these parameters).*

**R2-R1:** Corrected according to reviewer's suggestion. The sentence has been moved and rewritten in **section 3.4**. paragraph 5. Now it reads **"However, it has been described traditionally used sieve fractions method is considered unreliable because of the effect of morphometric parameters on the foraminifera weight (Beer et al., 2010).**

*R2-C1: Paragraph 3:*

*It is missing (and is instead necessary) a paragraph that deals with the **ecology** of the species that have been chosen for this type of analysis. Above all because calcification is a highly specific character depending on the ecology and living depth so I think it is necessary to analyse the ecology to understand if it can interfere in some way with the analyses proposed in this work.*

**R2-C2**: Authors agree with this suggestion. Now, **section 3.3**. in the material and methods describes the ecology of the 3 species following the bibliography provided by referee#2 and some additional sources. **It reads: "Ecology and life cycle of *G. bulloides*, *N. incompta* and *G. truncatulinoides"*.** The major ecological features of each species, as well as their life cycle has been described in order to clarify some points of the discussion. A clearer link has been established at **section 5.1**. when discussing the Optimum Growth Conditions.

*Technical corrections:*

**R2-C3:** *Line 133: It is not necessary to repeat the name of the genus, just bring it back pointed (G. bulloides). This inaccuracy is present many times in the text, pay attention on this.*

**Similar comments:** *Lines 232, 233: do not repeat the genus. Lines 232, 233: do not repeat the genus. Lines 822, 823: do not repeat the genus.*

**R2-R3:** Corrected according to suggestion. As some mistakes of the same type have been noted by reviewer#2, all of them have been corrected.

**R2-C4:** *Line 187: I don't know if it is a problem with the pdf that was generated by the automatic system but the references must be formatted like the rest of the text, now here have different font and size.*

**Similar comments:** *Line 286: Format as line 187. Line 757: Format as line 187.*

**R2-R4:** Corrected according to suggestion. This has been noted a few times and all mistakes have been corrected.

**R2-C5:** *Table 1: Format the last "bulloides".*

**R2-R5:** Corrected according to suggestion.

**R2-C6:** *Line 430: Seasonal and not Seasonnal.*

**R2-R6:** Corrected according to suggestion.

**R2-C7:** *References: In many cases, the species name is not given in italics in the references. I.e. Lines: 875, 917, 1009, 1021, 1061, 1084.*

**R2-R7:** Corrected according to suggestion.

REFERENCES:

Beer, C. J., Schiebel, R., and Wilson, P. A.: Technical Note: On methodologies for determining the size-normalised weight of planktic foraminifera, Biogeosciences, 7, 2193–2198, https://doi.org/10.5194/bg-7-2193-2010, 2010.

**AUTHORS COMMENTS ON ADDITIONAL CHANGES:**

**Comment 1.** There was a mistake in the previous version concerning the MOOSE program citing and the DYFAMED database references. This has been corrected in sections 3.1 and 3.4.

Now it reads: "**MOOSE program (Mediterranean Ocean Observing System for the Environment) (Coppola et al., 2019)" and "...DYFAMED database ([http://www.obs-vlfr.fr/dyfBase/index.php](http://www.obs-vlfr.fr/dyfBase/index.php)) (Coppola et al., 2021)**."

Authors would like to thank Laurent Coppola for his helpful comment concerning this issue.

**Comment 2**. The references list was not in the correct format in the previous version of the manuscript. This has been corrected to the Copernicus format.

REFERENCES:

Coppola, L., Raimbault, P.,Mortier, L., and Testor, P. Monitoring the environment in the northwestern Mediterranean Sea, *Eos, 100*, https://doi.org/10.1029/2019EO125951. 2019.

Coppola L., Diamond Riquier E. Carval T., Dyfamed observatory data. https://doi.org/10.17882/43749, 2021.

---

## Referee Report (RR1)

I have carefully read the revised version of the manuscript by Béjard et al. In general, I am satisfied with the changes made to the manuscript and think it is almost ready for acceptance. There is one issue, however, that troubles me and that I encourage the authors to critically look into their calculations. I suppose that if any changes are needed, they will not be of a nature that warrants revising the conclusions, but some of the numbers might have to be adjusted.

Like in the previous review, I remain concerned about the calculation of the flux-weighted mean shell weight of *N. incompta* shown in Fig. 6. The authors show a flux-weighted mean of approximately 3.3 micrograms (the red line in the figure below), which is supposedly calculated based on the monthly mean weight and the monthly mean shell flux (section 3.4). It makes sense to calculate the mean weight in this way, but I am unsure it is what has been done.

[Figure]

This is because  the monthly mean weights of *N. incompta* are always above 3.6 micrograms (see detail of Fig. 3 and 4 below).

[Figure]

So I don't understand how the flux-weighted mean shell mass can be below the monthly mean value. Either the authors have done the calculations in a different way, in which case

they need to update their method section, or there is an error and they need to carefully check their calculations. If there is an error, the authors should of course also double check the other calculations.

In addition, the ages provided in the legend of figure 6 are wrong for the trap (they should be negative for the BP ages) and, if I understand correctly what is shown, misleading for the sediments as these are only the ages of the dated samples, not of all samples analysed.

---

## Author Response (AR2)

First of all, authors would like to thank again referee #1 for the second review and the positive feedback on the reworked version of this manuscript. We acknowledge the time invested and are grateful for it.

Overall, we have focused this new version of the manuscript around the three recommendations suggested by the editor: "(1) the different observations from the traps and the sediments warrant a more careful interpretation and, (2), clearer information about the statistical modelling needs to be provided. Overall, the manuscript would benefit from a more focused discussion of significant patterns only".

Find here an abstract of the main changes that have been carried out on the manuscript and the answers to referee #1 second round of comments:

- Strong statements regarding the comparison between sediment trap and sediment cores (I.e.: strong calcification reduction, massive calcification loss, etc) have now been replaced by more conservative statements. Moreover, the possible role of selective dissolution as a possible factor impacting the calcification trends at the beginning of section 5.3 has been improved.
- Both the abstract and the conclusion now are more concise over the effect of environmental parameters on the different species calcification.
- GAM calculations details have now been included in section 3.6.
- Sections 5.1 and 5.2. are now more focused on significant patterns. Sentences that discussed or described non-significant trends have now been deleted as much as possible in order to keep the text lighter, specially around the chlorophyll-a and nutrients concentration effect on calcification, where a lot of non-significant trends were described. We also reduced and simplified as much as possible different sentences across the manuscript to keep it lighter.
- Some changes have been added in section 2 in order to better suit the map description.

**R1-C1: Difference between sediment trap and sediment shells.**
**The authors find that sedimentary shells are consistently heavier than shells in the sediment trap and attribute this (primarily) to a direct effect of ocean acidification, i.e. that shells in the sediment trap are less heavily calcified than their sedimentary (older) counterparts. Whilst this interpretation seems to make sense, it seems to go against the direct observations from the sediment trap, which show no clear relationship between calcification intensity and carbonate system parameters on seasonal time scales (Fig. 4) and no, or species-specific, response on interannual time scales (Fig. 5). So why would the three species respond in the same way if they show no, or different (even opposite in case of truncatulinoides) responses to more long-term change? It seems unlikely and is not explicitly explained in the manuscript. It is alluded to in the section about response and time scales, but in my opinion different observations from the traps and the sediments warrant a more careful interpretation and wording such as "reduction in shell calcification" (L1086) should be avoided.**

R1-R1: Authors agree. All the "strong statements" have been adapted and a paragraph at the end of section 5.3. has been added. It reads:" In summary, here we propose that a combination of a global scale process such as ocean acidification and a regionally amplified trend such as SST increase may be responsible for the MBW patterns differences between the pre-industrial and post-industrial to recent Holocene. However, the analysis of seasonal and interannual trends showed that the influence of the parameters is species-specific and varies across the time scale studied. This implies that other mechanisms may be affecting MBW trends and therefore, should be taken into consideration when interpretating these results."

**R1-C2: Additional modelling using GAMs.**
**It seems like these models have been sort of stuck on, rather than properly incorporated in the manuscript. Some of the results are even presented in the discussion, whereas they are perhaps better placed in the results.**
**That said, the main reason why I suggested GAMs was not just because it's a flexible method that allows to model non-linear relationships, but in order to model the calcification intensity as a function of seasonal and interannual variation (i.e. MBW = s(day of the year) + s(year), see e.g. https://fromthebottomoftheheap.net/2014/05/09/modelling-seasonal-data-with-gam/). GAMs are not the only method that can do this, my point was to model the calcification as a function of both seasonal and long-term variability. I don't think the authors have actually done this and hence remain unconvinced that the seasonal and interannual trends are statistically significant and consequently worth modelling further. So please provide clearer information about the statistical modelling rationale and approach and if two modelling approaches are warranted, integrate them better in the flow of the manuscript.**

R1-R2: Authors agree that the GAM description in the methods was short. We added more details to the GAM description in section 3.6. It reads: "The influence of a suite of environmental variables upon MBW$_{area}$ was assessed using General Additive Models (GAM) (fitted using the *gam* function from the *mgcv* R package). Due to data limitation, the GAMs could not be fitted to multiple independent variables, so potential effects of interacting environmental variables were could not be assessed. Each model tested the dependence of the different MBW upon a single independent variable: month or year, to evaluate seasonal and interannual trends; the flux of each species, to test effects of ecological variability; and a suite of environmental variables to determine impacts of various aspects of ocean chemistry on the calcification. Smooth functions of these measured quantities were used as the single independent variable within the GAMs, which were fitted using

the default settings of the *gam* function: a Gaussian family and identity link function; and the GCV.cp smoothing method. GAM results quantified the significance of the effect of each independent variable upon MBW." The GAMs have been modelled as suggested by reviewer #1, as a function od day/month/year… However, due to the lack of daily data concerning the calcification, a daily modelling was not considered as reliable, therefore the modelling has been done on a monthly basis, the highest resolution possible without losing to much data coverage.

Minor comments:

**R1-C3: L50 "likely to influence" make the abstract more informative and describe in which direction calcification is influenced, not just that it is. Similar remark e.g. for L57. Please check throughout the text (e.g. conclusions).**

R1-R3: Authors agree with the changes proposed here. The abstract has been updated and now reads: "The comparison of these patterns with environmental parameters revealed that calcification appeared to be species-specific". These changes have also been added in the remaining parts of the manuscript.

**R1-C4:L89: Hemleben et al seems an odd citation here.**

R1-R4: Authors agree. Citation replaced by Davis et al., 2017, Figuerola et al., 2021 and Orr et al., 2005.

**R1-C5:L129: this section comes out of the blue and stands on its own.**

R1-R5: Agreed. The sentence has been integrated in the next paragraph where the study zone is quickly introduced.

**R1-C6: L205: single sentence paragraph. Can this not be integrated better?**

R1-R6: Authors agree. Corrected according to suggestion.

**R1-C7: Table 1: perhaps replace "out of range" with "contains post-bomb material" or something like this. That makes it immediately clear what the age could be (out of range could also be too old).**

R1-R7: Authors agree. Now it reads: "bomb $^{14}$C".

**R1-C8: Section 3.3: I am surprised that the results of Rigual-Hernandez et al about the seasonality of the different foraminifera species is not mentioned here. It seems important for the optimal growth conditions later on. Consider Rebotim et al (Rebotim et al. 2017) for an extensive study of depth habitat and (Rebotim et al. 2019) for an analysis of calcification of truncatulinoides throughout the water column.**

R1-R8: Corrected according to suggestion. Now, seasonal abundances of each species are described at the end of each paragraph dealing with the ecology of the species. Also, the suggested citations have now been included.

**R1-C9: L290: odd citation for the presence of symbionts. Better use Takagi (Takagi et al. 2019).**

R1-R9: Citation updated.

**R1-C10: L334: perhaps add how much weight is lost in relative terms.**

R1-R10: Authors agree. Corrected according to suggestion.

**R1-C11:L633: Fig. 5 instead? Is Fig. 4 mentioned above?**

R1-R11: It is Figs 3 and 5 actually. Fig. 4 was not cited above, this has been corrected.

**R1-C12: L778-781: Since reproduction means death for planktonic foraminifera, the peak in the shell flux (dead organisms) is more likely to indicate the time of reproduction. Why is the migration to shallower water needed? I like the previous inferred energy allocation part, why is that not fine as an explanation? It could be discussed in the framework of the optimum growth conditions.**

R1-R12: The sentence about the migration has been removed. It was originally discussed in order to complete the energy allocation theory, explore the reproduction theory and expand the ecology part of the discussion. Finally, considering the referee #1 comments, the sentence around the water-column migration has been deleted.

**R1-C13: L816: since ch-a concentrations from remote sensing only pertain to the surface layer it would be better to write it like that and not make statements about the entire photic zone, which extends vertically.**

R1-R13: Authors agree. A sentence has been added to clarify this, it reads: "Also, note that the chlorophyll-a data presented here only represents the conditions in the surface layer."

**R1-C14: Fig. 4: why not show the results of the GAMs here in and in Fig 5?**

R1-R14: Authors see the point made here. The reason why the GAMs have been put in the supplementary methods is to avoid having complex statistical figures in the text and to promote figures in which recent trends are easier to observe on recent years. Also, with the new reworking of the manuscript, the focus has been put on reducing the discussion and text around non significant trends, and were are not sure including the GAMs would suit that strategy. However, if considered necessary, we would include the GAMs figures in the article.

**R1-C15: L1064: "since the industrial era and/or the late to recent Holocene". Still true?**

R1-R15: Authors see the confusion. Yes, it is still true, but the sentence has been reworded. Now it reads: "pre-industrial times to post-industrial and recent Holocene."

**R1-C16: Fig. 6: are the flux-weighted average values for incompta correct? They suggest really high fluxes for light shells, something I don't see in the figure in the supplement.**

R1-R16: The authors acknowledge that the flux-weighted values for incompta look odd, but the values and the calculations have been double checked and they look correct.

**R1-C17: L1199: I don't doubt the existence of multiple reproductive strategies in planktonic foraminifera, but it's unclear to me how this could influence the calcification intensity. I don't think this has been demonstrated, or suggested by anyone. Why end the manuscript like this?**

R1-R17: Authors understand the comment. The original idea was to explore potential causes of calcification variations that were not considered in our data, but it is true that, for the moment, there is not a study that proves the calcification differences between the reproductive strategies. Therefore, the sentence and references have been removed to avoid confusion.

---

## Author Response (AR3)

**Author's response:**

Find here the response to the 2 changes suggested by both reviewer #1 and the editor:

*(1) How exactly did you calculate the flux-weighted mean shell weight of N. incompta?*

The flux-weighted value has been calculated as described in section 3.4. The mean monthly flux for every month has been multiplied by the mean shell weight for every corresponding month and then divided by the total annual flux of the corresponding species. Reviewer #1 suggested to carefully check the values as they seemed odd *for N. incompta*. After checking, we found the mistake and the value of 3.3 µg is wrong. We apologize for the inconvenience. **The updated value is 3.9 µg**, which is more in accordance with the general weight of the sediment trap populations. This change has been done in all parts of the manuscript and the corresponding numbers and interpretations have been adapted.
Changes can be found in lines: 56, 627, 633, 641, 649, 965 and 986.

*(2) Please adjust the ages provided in the legend of figure 6.*

This has been corrected and now, the ages and the methodology is also referenced in the figure legend.
Changes can be found in Figure 6 and lines 922, 923 and 924.

On top of that, the manuscript has been carefully checked for grammar errors, writing mistakes and spelling.